# Designing meaningful continuous representations of T cell receptor sequences with deep generative models

Allen Y. Leary [1] ✉, Darius Scott [1], Namita T. Gupta [1], Janelle C. Waite [1], Dimitris Skokos [1], Gurinder S. Atwal[1] & Peter G. Hawkins [1] ✉

T Cell Receptor (TCR) antigen binding underlies a key mechanism of the adaptive immune response yet the vast diversity of TCRs and the complexity of protein interactions limits our ability to build useful low dimensional representations of TCRs. To address the current limitations in TCR analysis we develop a capacity-controlled disentangling variational autoencoder trained using a dataset of approximately 100 million TCR sequences, that we name TCR-VALID. We design TCR-VALID such that the model representations are low-dimensional, continuous, disentangled, and sufficiently informative to provide high-quality TCR sequence de novo generation. We thoroughly quantify these properties of the representations, providing a framework for future protein representation learning in low dimensions. The continuity of TCR-VALID representations allows fast and accurate TCR clustering and is benchmarked against other state-of-the-art TCR clustering tools and pre-trained language models.

T Cell Receptors (TCRs) are protein complexes that are present on the surface of T cells and are selected in the thymus to bind non-self peptide antigens[1]. This forms an important arm of our adaptive immune response with its ability to kill cells that have been infected or have other mutations that may cause harm, for example in cancer. Experimentally defined groups of TCRs with differing functional properties can lead to differing phenotypes, such as diverging cell states[2]. Tumor growth has been shown to be controlled in a way that correlates with the TCR function[3].

There are estimated to be $10^{15}$–$10^{61}$ possible human TCR proteins[4,5], with approximately $10^{11}$ T cells in an individual, there is consequently very limited overlap of TCR sequences between individuals. There is estimated to be an even larger space of possible antigens that these TCRs can bind, such that TCR cross-reactivity is thought to be essential[6]. This incredibly large state space with non-injective interactions make it challenging to build general predictive models for TCR-antigen interactions in an antigen agnostic manner.

It is becoming increasingly common to perform high-throughput measurements of TCRs in biological samples[7], but such data do not often include information about the antigens those TCRs bind. Though concurrent high-throughput measurements of TCRs and cognate antigens are possible[8,9] they require prior selection of antigens for study, leading to bias.It is therefore important to build computational tools to aid in clustering TCRs by functional properties, or by directly predicting the antigen recognition of TCRs.

Clustering TCRs that are sufficiently similar to bind the same antigen has been studied in the case where the antigen is known[8]. In the antigen agnostic case, TCR clustering has been approached via sequence-based evolutionary distance metrics between TCRs[10,11], and more recently via autoencoder[12,13] and masked language models[10,11]. The results of TCR-antigen screening can be used to train accurate supervised classification models of TCR-antigen interactions (see, e.g. refs. 8,13) and increasingly the performance of TCR-antigen classification models beyond the scope of those studied antigens (out-of-distribution), often by using both TCR and antigen information, has been studied[14–18]. This was recently extended to a zero-shot setting for predicting TCR-antigen interactions for TCRs and antigens that were not seen in the training data[17].

[1]Regeneron Pharmaceuticals Inc., 777 Old Saw Mill River Road, Tarrytown, NY 10591, USA. ✉e-mail: allen.leary@regeneron.com; peter.hawkins@regeneron.com

It would be desirable to have an atlas of TCRs that is able to cluster TCRs in an antigen-agnostic manner, classify TCRs by known ability to bind certain antigens, and to do so in an interpretable way. Further, with the advent of TCR based therapeutics[19] and the similarity of TCRs to antibodies, such an atlas should be generative such that the atlas provides a future route to de novo TCR/antibody design. Functional data for antibodies and TCRs are often limited and costly to perform, which makes Bayesian optimization a useful tool in this space to select sequences for further design iterations[20]. Bayesian optimization requires a low dimensional space in which a continuous function is optimized[21], and for that reason Bayesian optimization in the latent space of deep autoencoders (DAEs) has gained recent interest for complex data types[22–25]. We therefore set out with the following desiderata for a TCR atlas: (i) low-dimensional, (ii) interpretable, (iii) generative, (iv) smooth, (v) capable of clustering and classifying TCRs.

Masked language models of proteins and TCRs typically use large dimensional representations per amino acid[10,26] which can be problematic for representations of full protein sequences[27], and are not inherently generational in contrast with DAEs and auto-regressive language models such as GPT-3[28]. Although DAE representations of TCRs have been developed[12,13], the continuity of their latent spaces has not been throughly investigated. Indeed, investigation of the continuity of latent spaces in general is often only studied on toy models with known generative processes[29]. Disentangled Representation Learning (DRL) aims to learn representations of high dimensional data that improve the interpretability of models and their generational capabilities. DRL approaches have been applied to TCRs[30,31] but none to our knowledge have fully explored the tradeoffs between the landscape smoothness, interpretability, and sequence generation.

Here we train TCR-VALID (T Cell Receptor - Variational Autoencoder Landscape for Interpretable Disentangling), a capacity-controlled $\mathcal{C}\beta$-Variational AutoEncoder (VAE)[32,33] model trained using approximately 100 million unique TCRs from a combination of $\alpha$ and $\beta$ chains. TCR-VALID is built with low-dimensional representations, the disentanglement of which we quantitatively evaluate. We modify continuity metrics from machine learning literature to our biological context to measure the continuity of these representations in a systematic way. We benchmark TCR clustering and classification against other tools, showing that TCRs with similar sequences embedded closely in representational space provides state of the art TCR clustering. TCR-VALID's low dimensional, continuous, space provides a future route to Bayesian optimization of TCRs in addition to its clustering capabilities.

## Results
### Unsupervised learning of a TCR landscape via physicochemical feature embedding

Though the state space of TCR-antigen interactions is very large and we have very few measured interaction pairs involving a small subset of antigens, we do have larger volumes of TCR sequence data in the absence of known antigen pairings. We therefore hypothesized that using the unlabeled TCR sequence data to build latent representations of TCRs could then be used for downstream clustering, classification, and de novo generation. TCRs are made of two subunits, which for the majority of TCRs are $\alpha$ and $\beta$ TCR chains. Although these chains occur in distinct pairs, single cell sequencing is typically required to resolve the pairing and thus the datasets of paired TCR chains are much smaller than datasets of independent $\alpha$ and $\beta$ chains. Consequently, we chose to model TRA and TRB sequences independently in order to get a larger sampling of the state space of each sequence type during training.

The interaction of TCRs and antigen is primarily encoded by the Complementarity Determining Regions (CDRs)[1,34] of the TCR chains. The first two CDRs, CDR1 and CDR2, are encoded uniquely by a given V gene, whereas the CDR3 region occurs at the site of V(D)J recombination[1] and includes quasi-random nucleotide insertions and deletions. This leads to TCR chains being the product of a sparse discrete space for the V gene selection and a denser discrete space for the CDR3. This type of problem occurs often in biological sequence data where there is a sparse family structure that is complicated by dense variation at the sub-family level.

TCRs that bind the same antigen show biases to certain V gene usage[8], and so it is essential that information about V gene is encoded into the latent space for TCR clustering and classification. We sought to build a latent space that encodes the sparse discrete V genes and the highly variable CDR3 sequences in the same space such that we can co-cluster TCRs with similar yet distinct V genes, thereby precluding conditional-VAE models. The V gene can almost be uniquely encoded by the CDR2 sequence, so we chose to use the amino acid sequence of the CDR2 joined by a gap character to the the CDR3 to describe a single TCR chain (Fig. 1a). This is beneficial for encoding the biophysical similarity of two V genes, rather than relying on a dictionary embedding strategy[13]. The structural interaction of the CDR loops with the antigen is dictated by local physical interactions driven by the physicochemical properties of the amino acids of the loops. To capture this, we encoded the amino acids into 7 physicochemical features (see methods) forming a 2-D description for each TCR, and sought to learn smooth, low dimensional representations of these physicochemically encoded TCRs. We find that this physicochemical featurization of TCR sequences, at baseline slightly improves our ability to cluster TCRs over one hot encoding (see Supplementary Fig. 7) with the added benefit of improved interpretability and dimension reduction.

We chose to base our TCR-VALID architecture (Fig. 1a) on a capacity-controlled disentangling $\mathcal{C}\beta$-VAE[33], using a dataset of approximately 100 million TCR sequences (methods: data). Given our reasonably short sequences, we were able to make use of light-weight convolutional neural networks (CNNs) for the encoder and decoder to learn the highly non linear patterns that underlie the the key features of a given TCR. We explored the properties of the latent representations of TCRs (Fig. 1c) and the ability of the decoder to generate physicochemical representations of TCRs in a continuous space which ultimately form position weight matrices (PWMs) of TCRs (Fig. 1b, methods).

### Balancing landscape continuity with sequence reconstruction accuracy

Our VAE models on the physicochemical representations of TCRs generate a continuous space of physicochemical representations, and since these representations can be converted to PWMs (methods) these PWMs are themselves continuous and not discrete. This allows for small motions in latent space to slowly change the properties of the generated TCR sequence rather than discretely change specific amino acids. However, the sparse structure of the TCR space driven by the relatively small number of V genes can generate large regions of discontinuity in the latent space. This can cause challenges when clustering TCRs which may be similar in their CDR3 and V gene physicochemical properties, yet are distinct entities. It can also cause challenges in generation of TCRs due to regions of the latent space that do not correspond to the true manifold of physicochemically feasible TCRs.

For $\beta$-VAEs, one tunes the relative weight of the reconstruction loss and Kullback-Leibler divergence that forces the multivariate normal distribution on the latent space, where $\beta=1$ corresponds to a standard VAE. Importantly, due to the TCR physicochemical representations being continuous, one cannot measure the exact bits of difference between the input and reconstruction, meaning that $\beta=1$ does not carry the same meaning as in a typical VAE (see e.g. ref. 35). The capacity controlled VAE[33] allows one to control the quantity of information in the latent space via the capacity term, $\mathcal{C}$. This allows us to tune the smoothness of the latent space in a principled way, larger $\mathcal{C}$

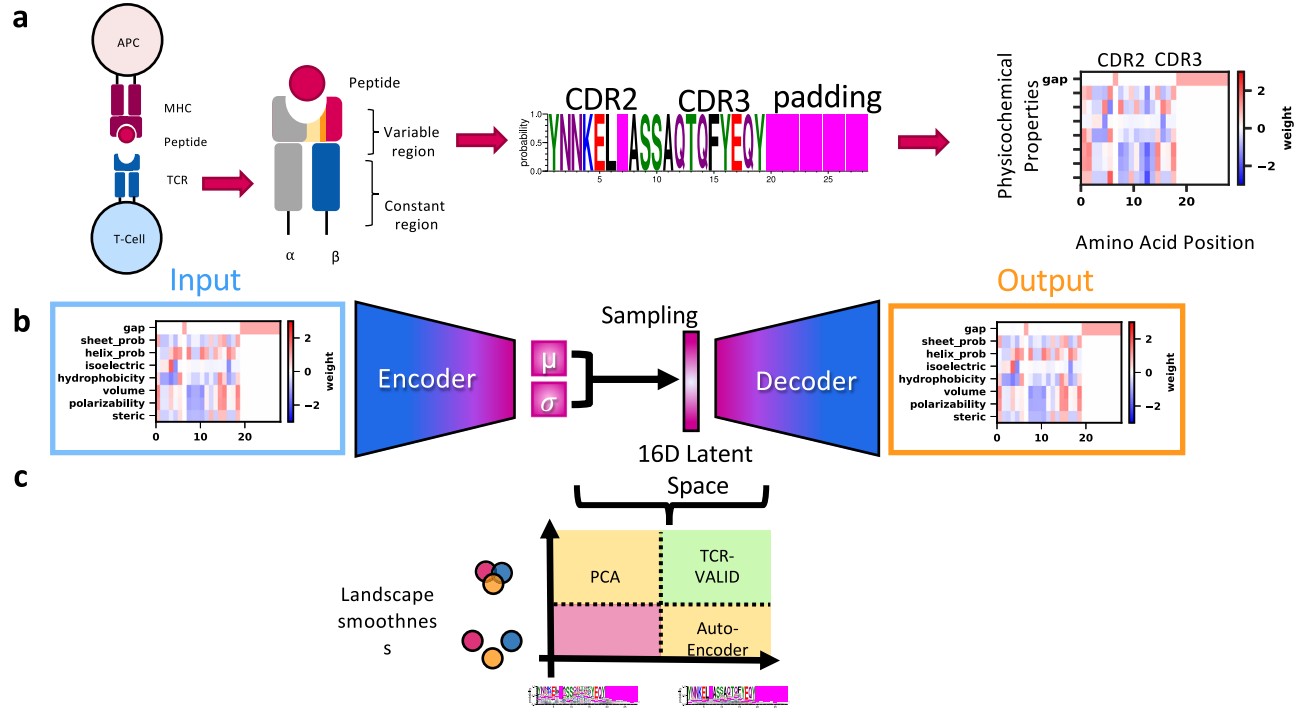

**Fig. 1 | TCR-VALID: Learning a physicochemically informed latent landscape for TCR sequences.** TCRs interact with peptide:MHC antigens on Antigen Presenting Cells (APC) primarily via the CDR loops of their variable region **a**, since the V gene usage can be encoded almost uniquely via its CDR2 and remaining diversity is in the CDR3, we use these sequence regions to encode a single TCR chain, and subsequently use physicochemical features of the amino acids to represent the sequence. **b** TCR-VALID architecture diagram: physicochemically encoded TCR sequences are used as input to train a $\mathcal{C}\beta$VAE to learn a continuous 16D latent representation of TCR sequences. **c** Diagram illustrating the tradeoffs between representation learning approaches for TCR sequences. TCR-VALID aims to balance those tradeoffs in order to provide learned landscape smoothness, interpretability without compromising TCR sequence reconstruction quality.

leading to better reconstruction at the expense of landscape continuity (Fig. 2a). We found this capacity limiting approach aids in preventing posterior collapse in the low $\mathcal{C}$ regime. The capacity term is simpler to implement and interpret than other methods of controlling the information in the latent space that rely on dynamically tuning the terms of the training loss[36,37].

We sought to quantitatively evaluate the continuity of latent TCR landscapes as a function of the information stored in the latent space. We aimed for all points between any pair of TCRs in the latent landscape to smoothly represent the manifold of feasible TCRs, driven by both discrete V genes and dense CDR3 sequences. Inspired by the latent transversals employed to evaluate simple image datasets[29,33], we developed a TCR latent transversal metric (methods). Briefly, we randomly select two unique TCR sequences with identical CDR3s and different CDR2s and embed them into our latent landscape with our trained encoder, and linearly interpolate between those two embedded TCRs in the latent space generating TCR PWMs along the trajectory (Fig. 2a). The distance $D_{cdr2}$ measures the difference between the CDR2 of the generated TCR and the closest observed CDR2 in the reference library, thereby measuring proximity to the true data manifold such that increases in this distance along a traversal indicate discontinuities in the latent landscape. Additionally we measure $D_{cdr3}$, which is the distance from the interpolated decoded CDR3 along the traversal to the CDR3 of the two endpoint TCRs. In order to not penalize models with lower ability to encode information due to lower capacity term $\mathcal{C}$, which will systematically have worse reconstruction accuracy even at the traversal endpoints, we normalize $D_{cdr3}$ to the distances endpoints (details in methods). We do not expect CDR3 to change along the trajectory and thus expect $D_{cdr3}$ to be small for a smoothly encoded latent space.

In order to balance the landscape smoothness with the amount of information the $\mathcal{C}\beta$-VAE latent landscape can encode we tuned the capacity $\mathcal{C}$ on a randomly sampled reduced dataset of ~4 million TRB chains. For the lowest $\mathcal{C}$ of 1 nat per latent dimension, we required a greater weighting of the capacity control term $\beta$ in the loss to reach the capacity (methods). Analyzing the average distances $\overline{D}_{cdr2}$ and $\overline{D}_{cdr3}$ over many Monte Carlo selected latent space traversals for a range of latent space information capacities we find that capacity of 2 or fewer nats per dimension leads to smooth, continuous latent representations (Fig. 2d). By visualizing the latent trajectories for an autoencoder and a hyperparameter-optimized TCR-VALID model with capacity of 2 nats per dimension, we can see that the learned latent landscape strays considerably further from the manifold of physicochemically feasible TCRs for autoencoders (Fig. 2b) as is reflected in the $D_{cdr2}$ and $D_{cdr3}$ metrics.

## Latent dimensions are disentangled and allow for principled TCR generation

Another key objective of TCR-VALID is to provide an interpretable TCR landscape which would enable characterization of the manifold of TCR sequences. Interpretability of TCR latent space can be defined as the ability to encode biological and physicochemical intuition and reasonability into our model output[38]. Unlike many datasets used to quantitatively benchmark disentangling properties, not all the ground truth generative factors for TCR sequences are known, labeled, or independent.

TCRs are generated biologically via the process of V(D)J recombination, wherein V and J genes are recombined with nucleotides deleted from their ends and quasi-random non-germline encoded nucleotides inserted at the joining sites. In $\beta$ chains, a D gene is additionally inserted between V and J, but hard to align due to its short

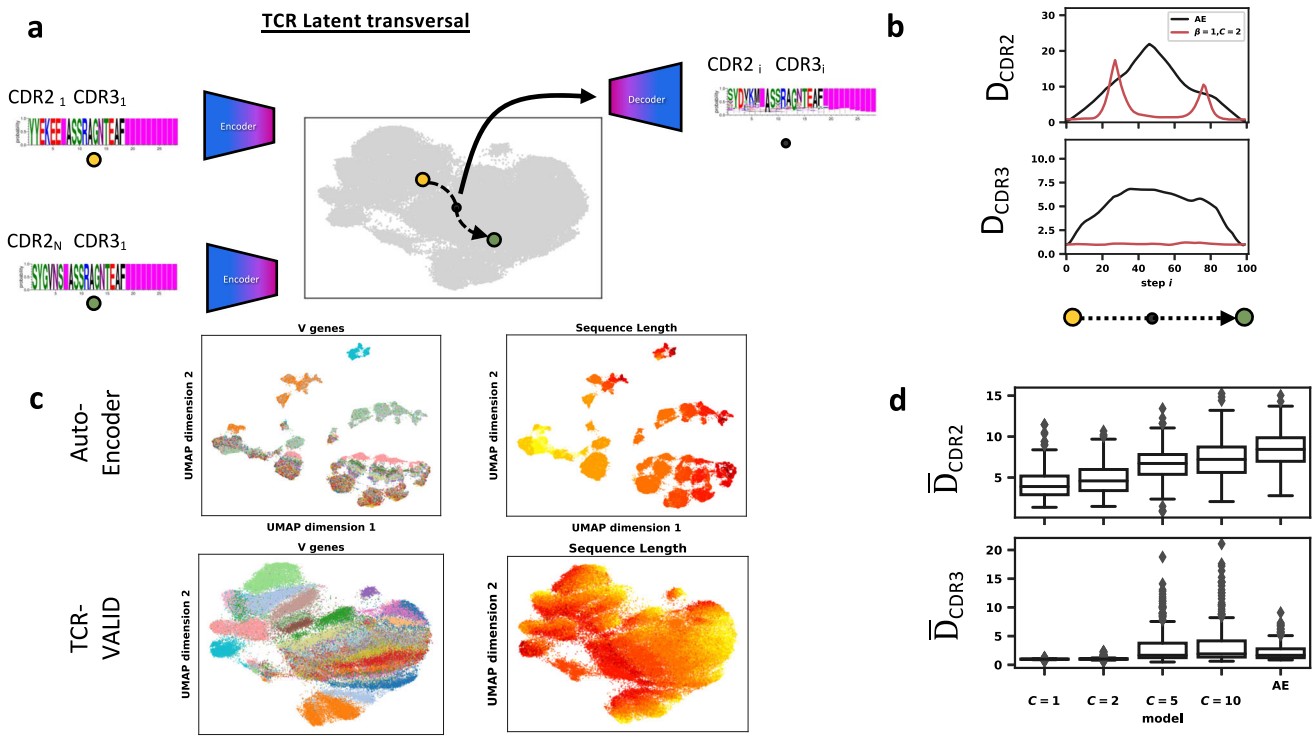

**Fig. 2 | TCR-VALID learns a smooth and realistic embedding of TCRs.**
**a** Illustration of TCR latent transversals to evaluate embedding landscape smoothness: two TCR sequences (CDR2-3) with identical CDR3 but differing CDR2 are embedded into the learned latent space. A linear interpolation between TCRs in latent space is then decoded back into sequence space and evaluated. **b** The distance between TCR-VALID (red) and auto-encoder (black) decoded interpolated TCRs and training TCR sequences for CDR2 ($D_{cdr2}$ metric, top) and CDR3 ($D_{cdr3}$, bottom) $n = 100$ interpolated points per trajectory, **c** UMAP embedding for subset of TCR training sequences for auto-encoder (top) and TCR-VALID (bottom) colored by V gene usage (left panel) and sequence length (right panel). **d** Averaged distance over the whole trajectory for many Monte Carlo selected latent space transversals for both CDR2 ($\overline{D}_{cdr2}$ metric, top) and CDR3 ($\overline{D}_{cdr3}$ metric, top). Box plots are shown with lines for quartiles, whiskers extend to 1.5 times the interquartile range, and outliers above or below the whiskers are displayed as points. $n = 310$ traversals per boxplot. Source data for **b**–**d** are provided as a Source Data file.

length and the CDR3 deletions. We hypothesized that the latent space of TCR-VALID could be able to disentangle V and J gene usage and the mean physicochemical properties of the quasi-random insert region between the V and J genes. Given that the training objectives of TCR-VALID do not explicitly aim to disentangle these predetermined generative factors they likely encode factors beyond those we study, particularly properties relating to the non-mean properties of the insert region.

A well disentangled representation would encode distinct generative factors in non-overlapping subsets of dimensions. To quantitatively benchmark TCR-VALID's performance and tune its hyper-parameters, we employed a disentangling score[39] that has been identified as robust and broadly applicable[40]. Importantly, this scoring scheme accounts for generative factors that are unknown, but still encoded within the latent space. Briefly this scoring scheme measures the importance of individual latent dimensions for predicting exclusively a single generative factor combined with a weight term accounting for total contribution of the latent term to generative factor prediction (details in methods).

We trained random forest (RF) models[41] on the previously trained $\mathcal{C}\beta$-VAE latent representations from the sub-sampled TRB training set with a range of capacities $\mathcal{C}$ to predict the three key TCR generative factors for the associated TCRs (Fig. 3a, methods). The feature importances can then be used to both score the disentanglement (Fig. 3d) and be visualized via Hinton diagrams showing which dimensions encode differing generative factors and how strongly (Fig. 3b). TCRs projected into the latent dimensions for TCR-VALID ($\mathcal{C} = 2$) that encode primarily V gene or J gene usage show clear

structural stratification by V gene in these dimensions without any further dimension reduction (Fig. 3c, top), and those same features are mixed in latent dimensions not identified as encoding V/J genes and encode still unknown TCR factors (Fig. 3c, bottom). The TCR-VALID model that displays both the highest disentangling score and lowest reconstruction loss was the same model that presented the smoothest learned landscape with capacity of 2 nats per dimension. One potential flaw in the disentangling score, is that it aggregates all generative factors into a single score[40]. We found via hyper-parameter tuning that capacity 2 gives the best balance between disentangling score and sequence reconstruction accuracy as captured by the reconstruction loss (Fig. 3d), whilst PCA and autoencoder models sacrifice reconstruction accuracy or latent landscape disentangling.

## TCR-VALID performs fast and accurate antigen specific TCR clustering
TCR sequences that are projected closely into the latent landscape of TCR-VALID have closely related physicochemical features which might underlie their binding mechanisms to the same antigen peptides[42]. Along with the disentangling and interpretability properties of TCR-VALID, we sought to benchmark its TCR-antigen clustering capabilities against current state of the art approaches, some of which were developed for the sole purpose of TCR-antigen pair clustering. We fixed $\mathcal{C} = 2$, determined to give us the optimal smoothness and disentangling properties and trained a TCR-VALID model on the full training dataset of approximately 100 million unique TRA and TRB chains (methods: data) in order to evaluate its TCR-Antigen clustering capabilities.

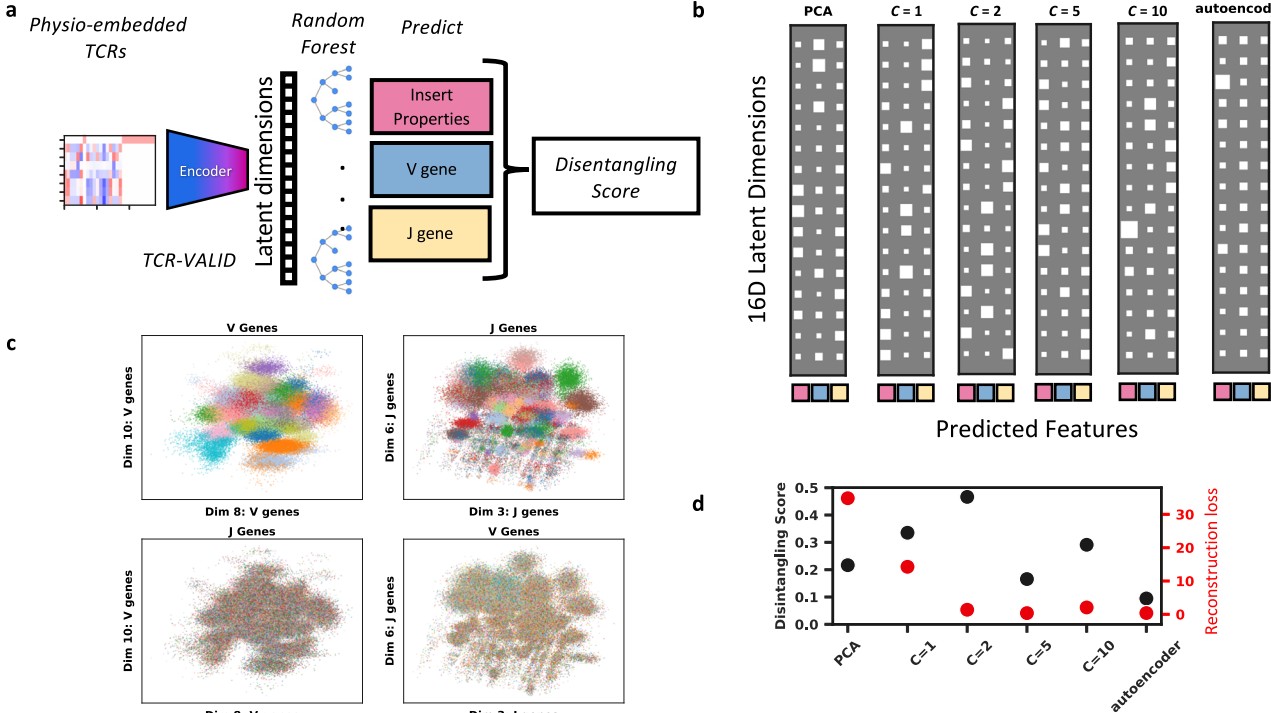

**Fig. 3 | TCR-VALID learns a disentangled interpretable latent landscape.**
**a** Quantitatively evaluating the disentangling of TCR-VALID by training random forests to predict TCR-intrinsic factors in the learned latent dimension and computing a Disentangling Score[39] on their feature importances. **b** Hinton diagrams illustrating the relative importance of latent dimensions from trained networks at predicting TCR intrinsic features. Square size indicates relative importance, rows indicate latent dimensions and columns indicate respectively mean insert physicochemical properties, V gene and J gene. **c** Plotting the learned latent landscapes with the feature they best encode (top) show a smooth yet disentangled landscape for V (left) and J (right) genes. If we plot the none encoded feature in the inappropriate dimensions the organization is lost (bottom), **d** Disentangling score versus reconstruction loss for TCR-VALID hyperparameter range versus PCA and autoencoder approaches. Source data for **b**–**d** are provided as a Source Data file.

We used a comprehensive labeled TCR-antigen dataset (methods) of high quality paired-chain TCRs that bind a wide range of antigens to evaluate our model's generalizability.

Given that most TCRs cannot easily be clustered, evaluating TCR clustering performance must account for both the quality of the TCR-antigen clusters produced and the ability to capture many different TCR-antigen interactions. Our benchmarking uses two parameters which are very similar to those used in other works[43,44] (Fig. 4a, methods): clustering precision that measures the percentage of clustered TCRs in 'pure' clusters (methods) and a clustering Critical Success Index (CSI) that measures the percentage of all TCRs that are in 'pure' clusters. The comparison in Fig. 4 shows performance with a minimal cluster size of 3 unique TCRs; performance with a minimal cluster size of 2 is shown in Supplementary Figs. 2–6. Another confounder for TCR clustering tool comparisons is that different TCR clustering tools use only TRB, TRA, or paired chain information, and some tools use only the CDR3 or some subset of the CDRs. In order to compare the most tools possible, we focus on TRB clustering with the most CDR regions possible. We also compared varying combinations of TCR chain and CDR usage in Supplementary Fig. 1.

Most TCR clustering tools have some notion of a clustering radius, that broadly determines how far apart TCRs can be for TCRs to be co-clustered. For a small radius few TCRs are clustered but those that are are placed in high quality clusters (high precision, low CSI). As the radius increases clusters get larger and greater in number such that CSI increases at some cost in precision, until eventually precision and CSI drop to zero. It is therefore only fair to compare tools in both precision and CSI, and ideally as a function of the tools' radius where one exists. For tools that create representations or distance graphs we either used their own radius where one existed ('threshold' for iSMART[45]), or used DBSCAN[46] with its distance parameter $\varepsilon$ (ESM[26], TCR-BERT[10], tcr-dist[47],

deepTCR[13]). For clusTCR and GLIPH2 we were unable to tune any effective radius, so we only tracked precision and CSI at default settings. Additionally clusTCR and GLIPH2 allowed TCRs to be placed in more than one cluster and this was adjusted as described in the methods to allow for a fairer comparison with other tools, though we investigated the effect of this correction across several datasets (Supplementary Figs. 2–6).

We find that without spike-in the TCR-VALID curve overlaps with sequence-based approaches designed in part by human guided feature selection such as tcr-dist[47] and iSMART[45] indicating similar performance along our key metrics of CSI and precision. The TCR-VALID curve is consistently above GLIPH2[43] and deepTCR[13], a deep learning model with larger latent space (Fig. 4c, left panel) indicating that it outperforms them on this pair of benchmarks. Further, we benchmarked TCR-VALID against recent general protein transformer-based models[26] and TCR specific transformer models[10] (Fig. 4c, left panel) that learn high dimensional embeddings of TCR sequences (methods) and found the curve of our approach is above these approaches.

In addition to profiling the clustering performance on the reference dataset, following the work of Huang et al.[43] we tested the robustness of the TCR clustering approaches by spiking in irrelevant TCRs from a CD4 TCR reference set[43]. We measured the decay in both CSI and precision for the top performing tools without spike-in (tcr-dist, iSMART,TCR-VALID), the fastest tool (clusTCR), and GLIPH2. We find that the performance of every TCR clustering tool decays upon increasing spike in fold of irrelevant TCRs (Fig. 4c & d) as indicated by the curves collapsing to 0. Our results align with the recent findings that many of the TCR clustering tools produce similar clusters[48] and that more attention must be paid to the strength of the use case for each tool. We find that some tools are very sensitive to the labeled

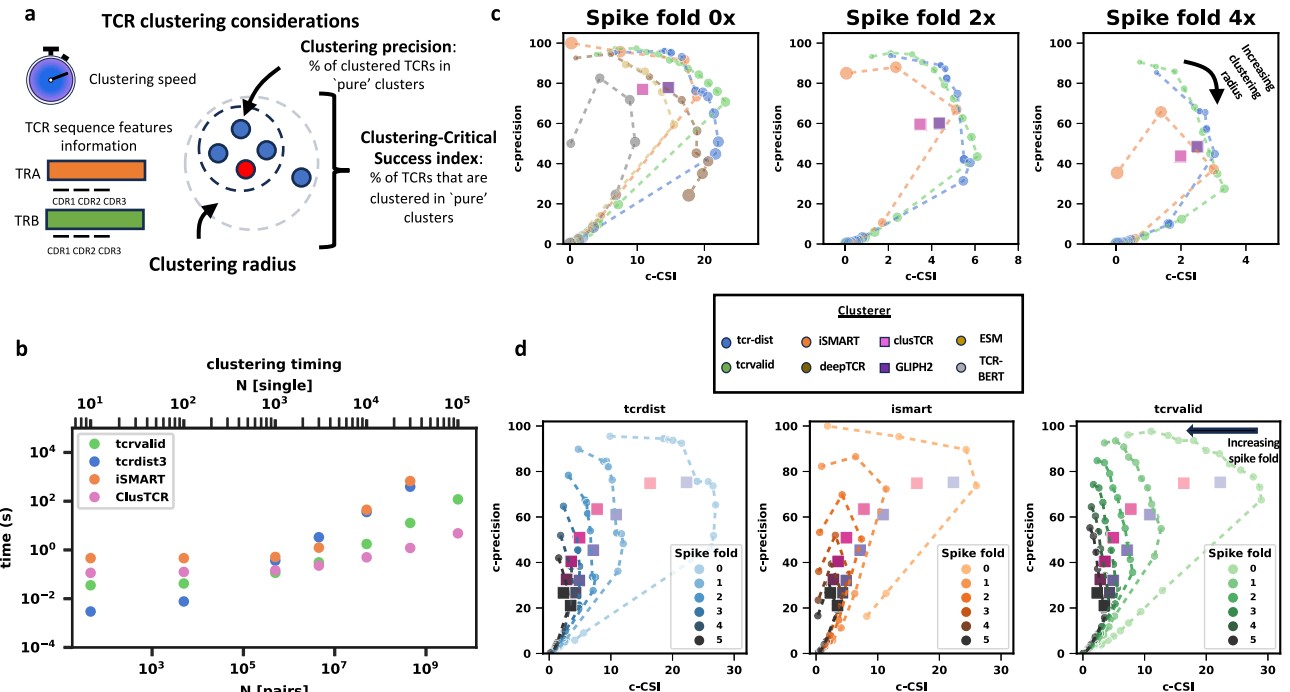

**Fig. 4 | TCR-VALID clusters TCR-antigen pairs quickly with state of the art performance. a** Illustration of considerations when comparing TCR clustering approaches (clockwise from top left): clustering speed, clustering precision (c-precision), clustering Critical Success Index (c-CSI), clustering radius/threshold, TCR sequence features. **b** Clustering time benchmarking of TCR-VALID versus other TCR clustering approaches as a function of number of sequences clustered. **c** Clustering precision versus Clustering-Critical Success Index (details in methods)

for TCR-VALID versus TCR clustering tools for increasing spike fold of irrelevant TCRs. TCR clustering curves represent increasing clustering radius from left to right. **d** Clustering precision versus Clustering Critical Success Index for tcr-dist, iSMART and tcrvalid for increasing spike folds (0–5×) illustrating the effect of irrelevant TCR spike in on the respective tools. Source data for **b**–**d** are provided as a Source Data file.

TCR-antigen reference they are trained or tested against (supplementary Fig. 2).

Since a common use case for TCR clustering is to identify clusters across multiple large TCR repertoires from different individuals it is important that the clustering algorithm scales well with data volume. We benchmarked the time complexity of TCR clustering algorithms, finding that TCR-VALID scales better than tcrdist3[49] and iSMART (Fig. 4b). clusTCR scales better than TCR-VALID but has lower clustering performance. clusTCR's speed is derived largely from the use of approximate nearest neighbor clustering methods, which may offer a performance boost over DBSCAN if used with the fixed size representations of TCR-VALID. TCR-VALID embedding speed places a lower bound on the clustering speed and this lower bound scales similarly to clusTCR (methods, Supplementary Fig. 8). We were unable to benchmark GIANA[50] due to its licensing restrictions, though Hudson et al.[48] benchmarked GIANA against other tools and found its speed was similar to iSMART and GLIPH2 and its clustering performance similar to clusTCR.

### Disentangled representation provides cluster interrogation and de novo generation within clusters

TCR-VALID's disentangled representation allows us to probe which dimensions, and associated quantities, independent clusters of TCRs that bind the same antigen differ. We find that the two largest clusters (large cluster $n = 363$, small cluster $n = 40$ unique sequence) we identify from with TCR-VALID bind HLA-A*02:01 GILGFVFTL (influenza) peptide (Fig. 5b, left panel).

We confirm that the two clusters share V gene usage (TRBV19) but differ in J gene usage. As expected, the insert region is variable both within a cluster and between clusters. As validation of the disentangling and interpretability of our learned latent landscape, we showed that in dimensions that encode mean insert physicochemical

properties (0,8) and J gene usage (12) the two clusters clearly segregate. Conversely, given that the clusters largely share V gene usage, along dimension 1 that encodes V gene we find the two clusters overlap (Fig. 5b, right panel). These results highlight the benefits of the interpretable learned landscape when relating TCR-antigen binding clusters in an unsupervised fashion to both recover known TCR features such as V,J gene usage and learn biophysical properties such as mean physicochemical insert value which underlie different binding mechanisms.

We demonstrate de novo TCR sequence design for a TCR associated with each of the two large flu clusters by using the mean of the TCR representation of the TCRs belonging to each cluster (Fig. 5a) and decoding these back into physicochemical property via the trained decoder. The decoder output of a physicochemical map is then converted into probability of amino acid identity (Fig. 5c, methods). These designed sequences display motifs that have previously been identified in similar datasets with a supervised approach[8].

### Latent representations provide universal feature extractor for TCR classification with uncertainty-estimation

TCR clustering is particularly useful in extracting putative groups of functionally similar TCRs in an antigen agnostic way, however there are some antigens for which many cognate TCRs are known. In those cases, we can build meaningful classification models by learning antigen specific non-linear mappings from the TCR sequences, rather than relying purely on proximity in a physicochemical latent space, see e.g., refs. 8,13. This can be particularly useful for TCRs that recognize common viral antigens, which can be clonally expanded in samples taken during an immune response. It can be useful to identify these to characterize response to infection or in the context of a tumor to understand the role of bystander (not tumor reactive) TCRs[51–53].

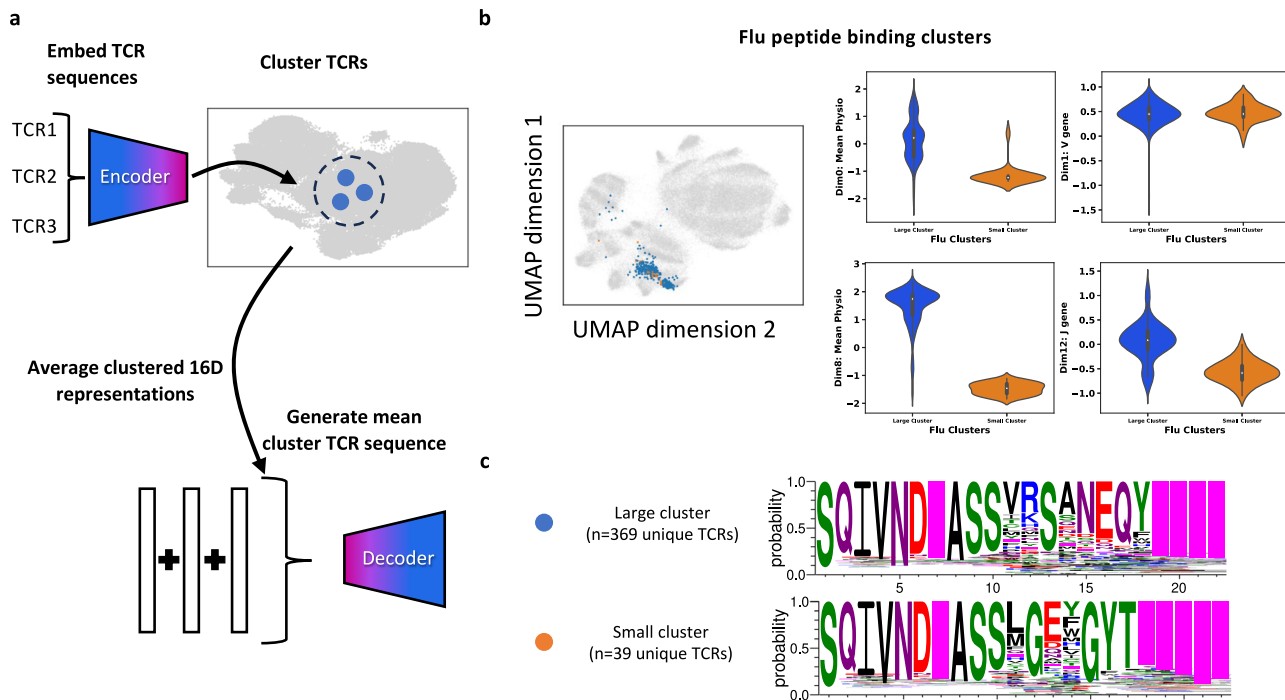

**Fig. 5 | TCR-VALID can be used to both identify TCR-Antigen binding modes and generate meta TCR sequences. a** Unique TCRs can be embedded and clustered in TCR-VALID latent space, their 16D representations can then be averaged to create meta representation of the TCR cluster. This meta representation can then be decoded to generate the meta-TCR sequence representative of the cluster. **b** TCRs from validated TCR-antigen pairs embedded into the learned latent landscape (left) and TCRs from the two largest flu clusters (Large cluster n = 369 unique TCRs and Small cluster n = 39 unique TCRs) embedded into learned latent

landscape. Violin plots of latent dimensions of TCR-VALID model for two flu clusters above illustrating separation of clusters along Mean Insert Physicochemical properties and J gene usage but overlapping for V gene usage (right). White dot, median. Box edges, 25th and 75th quartiles. Whiskers, 1.5 × the IQR of the box edge. **c** TCR PWMs generated from the mean TCR-VALID representations of the two largest flu clusters, illustrating conserved CDR3 motif and gene usage. Source data for **b** are provided as a Source Data file.

One problem with classification models in biological data settings is how the models behave on data that is out-of-distribution (OOD) as opposed to in-distribution (ID)[54,55]. In the case of TCR-pMHC interactions, models are trained on a limited subset of the large state-space of TCRs and antigens. A corollary of this is that models trained on TCRs binding a limited set of antigens are likely to perform poorly when evaluated on large repertoires of TCRs. This is because most TCRs are unlikely to bind any of the antigens associated with the training data. It is therefore important to either: (a) build models capable of predicting any TCR binding to any antigen; or (b) understand the confidence of a models prediction on both ID and OOD TCRs to limit the rate of confident predictions on OOD TCRs dwarfing the predictions of TCRs that truly bind antigens of interest.

Tackling question (a) has been approached by learning representations of both the TCR and peptide, using classifiers directly from the sequence or features of the sequence[16-18] or using autoencoders[14,15]. These models still underperform on OOD peptides[14-18], though Gao et al.[17] recently used a meta-learning approach to perform state-of-the-art TCR-pMHC binding prediction for OOD peptides. Due to the relatively small volume of TCR-pMHC data available, such strategies are still not expected to perform with complete accuracy in the full state-space of TCR-pMHC interactions. Therefore it is still important to consider model confidence in such strategies when predicting pMHC labels on large repertoires of TCRs. For this reason, and that we here focus on a low-dimensional TCR representation model, we instead focus only on question (b) in the context of models that classify pMHC labels using only TCR as input. Namely, how well do models' confidence on their predictions allow one to separate the regions of TCR space it knows about (in the region where its training data was, ID) from those it does not (where there was no training data, OOD).

Due to the low dimensionality and capacity limit of TCR-VALID, its TCR representations are densely encoded into the latent space with some loss in information of the representations from the original data. One may expect this leads to poor classification and OOD detection performance. We studied: i) classifier performance for ID antigen labels using TCR-VALID representations of ID TCRs, ii) classifier performance at distinguishing whether a TCR is ID or OOD, that is are the model's certainties in its predictions useful for evaluating whether to trust the model's predictions, and iii) can the performance at OOD detection be improved using unlabeled TCR data and their TCR-VALID representations.

We split the TCRs in the antigen labeled TCR dataset into two groups: those TCRs that bind an HLA*02 pMHC (ID) and those that don't (OOD). We then trained a small classifier to predict the antigen label of the HLA-A*02 associated TCRs in a multi-class setting (Fig. 6a). We evaluated the models' confidences in their predictions for ID and OOD TCRs (Fig. 6b). We evaluate a model's OOD detection by finding the AUROC between the model's confidence (methods) on test TCRs from the ID TCRs and OOD TCRs. We compared our models performance in ID classification and OOD detection with a popular TCR classifier, deepTCR[13], in its classifier mode (Fig. 6c). DeepTCR trains directly on TCR sequences and genes and has a final representation size of 256 dimensions prior to the multi-layer-perceptron layers. This is in contrast with the pre-trained TCR-VALID model with representations having just 16 dimensions and with a constrained informational capacity.

To improve our OOD detection we utilize a portion of the unlabeled, random repertoire, TCRs and the method of Lee et al.[56] to add an auxiliary loss to force the classification probabilities to the antigen targets to be more uniform when samples come from the unlabeled set

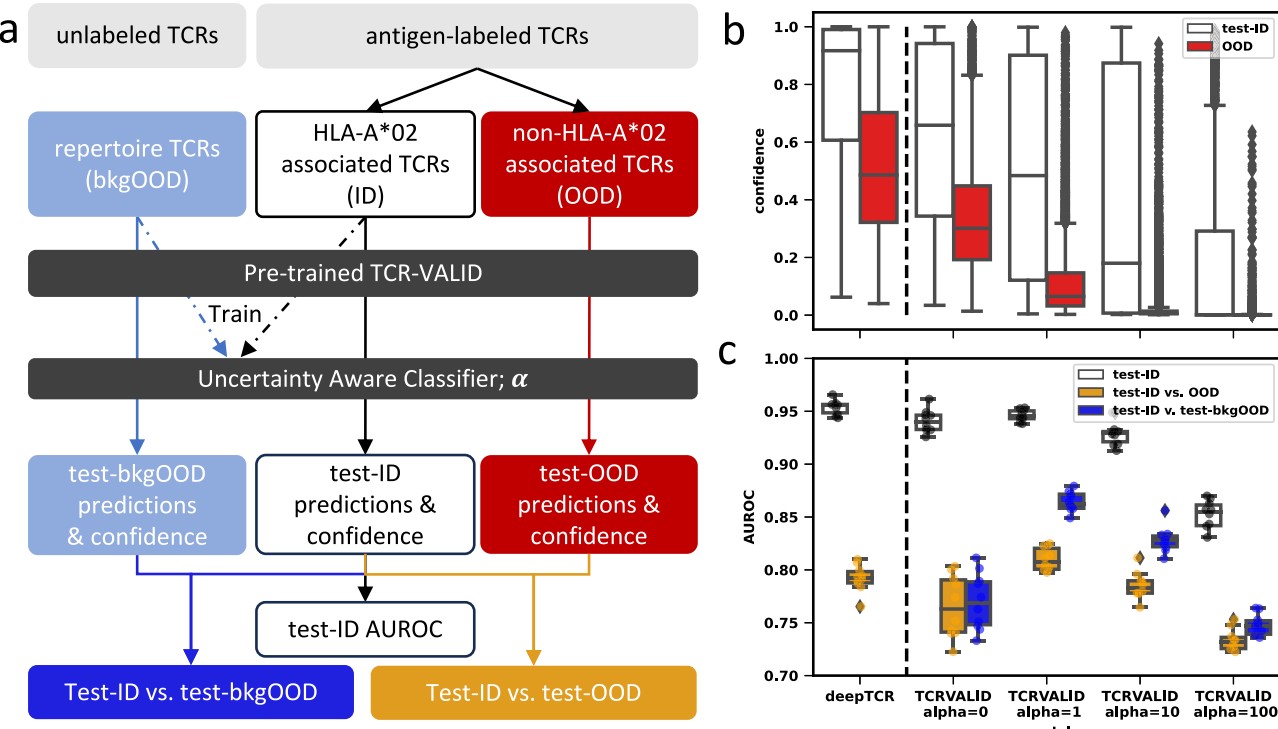

**Fig. 6 | TCR-VALID performs strong classification and OOD detection despite low dimensional representations and information bottleneck. a** schematic of the experimental set-up. labeled TCRs are divided into HLA-A*02- (ID) and non HLA-A*02 (OOD) associated groups of TCRs, and unlabeled TCRs (bkgOOD). An uncertainty aware classifier is trained on representations from the pre-trained TCR-VALID model. The classifier has the joint task (weighted by $\alpha$) of classifying which ID peptide the TCR binds and making predictions more uniform for bkgOOD TCRs. The model is evaluated on the classification task, and the ability to distinguish ID and OOD (bkgOOD, or OOD) TCRs based on the model confidence. **b** Model confidence for TCR-VALID with various parameters $\alpha$, and for DeepTCR. Model confidence is calculated for the test data of the ID TCRs (test-ID; $n = 3530$, ten 12.5%

Monte Carlo test sets of 2817 TCRs; $n = 3420$ for DeepTCR due to internal removal of 93 TCRs) and the OOD TCRs ($n = 8760$; confidence on 876 TCRs for models trained on each of ten folds). **c** AUROC displayed as box-plot over ten Monte-Carlo cross-validation splits for the three tasks: classification of ID TCRs (test-ID), OOD detection (test-ID vs OOD) and bkgOOD detection (test-ID vs test-bkg-OOD). Increasing $\alpha$ from 0 to 1 has no statistical change in ID classification (median AUROC 0.940 vs 0.945, two-sided Mann–Whitney $U = 29$, $p = 0.06$, $n_1 = n_2 = 10$), but large improvements in OOD detection for both bkgOOD and OOD TCRs. Box plots are shown with lines for quartiles, whiskers extend to 1.5 times the interquartile range, and outliers above or below the whiskers are displayed as points. Source data for b,c are provided as a Source Data file.

(methods). These loss terms are balanced by a parameter $\alpha$. Tuning $\alpha$ from zero, where there is no OOD detection improvement included, to larger values of $\alpha$ where the output distribution is progressively forced to be uniform for unlabeled data.

We find that while the auxiliary task is trained only on random repertoire TCRs, OOD detection improves not only for random TCRs but also for the non-HLA-A*02 TCRs for $\alpha = 1$ compared with $\alpha = 0$ (median AUROC 0.808 vs 0.763, two-sided Mann–Whitney $U = 7$, $p = 6.6\mathrm{e}{-4}$, two-tailed, $n_1 = n_2 = 10$) with no appreciable drop in classification performance on the in-distribution task (median AUROC 0.940 vs 0.945, two-sided Mann–Whitney $U = 29$, $p = 0.06$). This process leads to TCR-VALID with $\alpha = 1$ performing OOD detection better than deepTCR (median AUROC 0.808 vs 0.792, two-sided Mann–Whitney $U = 12$, $p = 2.3\mathrm{e}{-3}$) and with similar in-distribution classification to deepTCR, though with statistically lower performance (median AUROC 0.945 vs 0.956, two-sided Mann–Whitney $U = 15$, $p = 4.6\mathrm{e}{-3}$). This method for improving OOD detection using unlabeled TCRs could be applied to other TCR classification models, improving the informative predictive score of whether a new TCR of interest is ID for the model, aiding users identify potential false positives, particularly for repertoire TCRs which is a common use case.

## Discussion

In this study we present TCR-VALID, a $\mathcal{C}\beta$VAE trained using approximately 100 million unique TCR sequences to learn smooth, interpretable, and low-dimensional representations of TCRs with

generative ability. TCR-VALID is fast, lightweight, and provides state of the art clustering of TCRs by antigen binding properties via its representations without need for retraining, and these representations can be used for TCR classification and OOD detection.

One future use of TCR-VALID's disentangled TCR physicochemical landscape is expected to be for circumventing some of the current limitations in TCR repertoire profiling. Namely, an inability to relate unique yet similar TCRs seen across individuals. This is often the case in disease settings, where due to limited TCR overlap between individuals few TCRs are found shared between individuals despite a shared disease and therefore assumed TCR response. TCR-VALID provides fast and accurate clustering of functionally similar TCRs, allowing it to be quickly applied over many TCRs from large repertoires. Clusters of TCRs present in patients with shared disease, or under the same treatment, may provide a tool to reveal TCRs involved in those conditions. The disentangled space gives route to understand the conserved nature of those similar TCRs.

Other auto-encoding frameworks have been applied to TCR data[12–14,30]. They have previously either been: limited to antigen-labeled TCR data thereby limiting the scope of TCR sequence diversity captured in the representations; have not investigated the disentangled nature in a quantitative manner; or, have not investigated the clustering capabilities within the latent space. We applied robust validated metrics from the DRL literature[40] and quantitatively assessed our learned latent space smoothness, to optimally choose the information bottleneck of TCR-VALID. To the best of our knowledge this is the most

comprehensive investigation of TCR sequence latent space using the quantitative tools of DRL.

Other approaches have shown promise in the DRL literature such as FactorVAE[57] and $\beta$-TCVAE[58] that further reinforce the disentangling objective of $\beta$-VAE by adding further terms to the training loss based on latent dimension correlations and an additional discriminator respectively. These tools may provide an increase to the disentangling abilities of TCR representations.

In this work we have shown that the key generative factors for TCR sequences can be well disentangled by a $\mathcal{C}\beta$ VAE in a small number of dimensions, with well quantified smoothness, and with good reconstruction accuracy. This smooth, low dimensional, generative space may open up possibilities for latent space optimization of TCR sequences, which would be an exciting avenue for future research.

# Methods

## Data

We collected two sets of TCR data, an unlabeled set of TCRs from repertoire-level data, that is without corresponding antigen-binding information, and a set of antigen-labeled TCRs for which the cognate antigen is known.

We collected repertoire level TCR data from the iReceptor Gateway[59] (https://gateway.ireceptor.org/login) and VDJServer[60] (https://www.vdjserver.org). These data include TRB and TRA chains, that are predominantly unpaired.

We collected paired-chain TCRs with known cognate antigens from two sources; those associated with[8] and VDJdb[61,62]. For VDJdb we collected all human paired-chain TCRs with a quality 'score' of at least 1 (accessed October 2021). In order to investigate the effect of TCR-antigen reference data sets for TCR clustering we equally benchmarked clustering tools with the TCR-antigen reference data set from ref. 43 before and after filtering for quality scores of at least 1 as mapped from VDJdb. To undertake irrelevant TCR spike-in benchmarking we followed the process set out in ref. 43, briefly we collected the same CD4 reference data set and created 10 replicates for each spike in fold ranging from 1-5× the GLIPH2 reference data set size $n = 3262$. These irrelevant TCRs were then each given a unique hashed cognate antigen value and then concatenated with the labeled TCR-antigen reference data sets for downstream scoring.

**Data preparation.** The ingested TCR data is in the AIRR schema (https://docs.airr-community.org/en/v1.2.1/datarep/rearrangements.html). We clean the ingested data to train the model on high-quality TCR sequences. The first step in the quality control pipeline is selecting the locus of the TCR, as either TRB or TRA, and chains that are True for 'productive' and 'vj_in_frame', and False in 'stop codon'. Several criteria were used to evaluate the quality of the junction sequences of the TCRs. The amino acid junction sequence must have a length greater than or equal to 7, a length less than 24. We required that the junction must start with the amino acid C, end with the amino acid F, and not contain X, *, or U. Very few sample TCRs have the CDR1 and CDR2 sequences labeled. However, a large percentage are labeled with the column named 'v_call', which gives the V gene that encodes for the CDR1 and CDR2 sequences. We thus use 'v_call' to annotate CDR1 and CDR2. To prevent ambiguous 'v_call's affecting the quality of CDR labels we restrict TCR chains to those for which the 'v_call' contains a single V gene while allowing for multiple alleles of a single V gene. After filtering ambiguous 'v_call's we assume all V alleles are *01 as the CDR2 sequence between alleles of the same V gene are almost always identical. To assign CDR1 and CDR2 to each chain we first retrieve the amino acid sequences for each of the TRV genes from (https://www.imgt.org/genedb/). This database provides sequences for all human TRV genes, including each allele, for each gene we retain only *01 alleles. For each sequence we use ANARCI[63] to apply IMGT numbering,

and retrieve the IMGT CDR1 and CDR2 regions for each sequence. These CDR1 and CDR2 definitions for each v gene are then joined into the dataset via the 'v_call'. TCR chains for which any of: 'v_sequence_end', 'cdr3_end', 'j_sequence_start', 'cdr3_sequence_end' are null are removed. We define the insert amino acid sequence as the (in-frame) codons in the cdr3 nucleotide sequence which are encoded by 1 or more non-V/J encoded nucleotide, as determined by AIRR sequence schema fields: 'v_sequence_end', 'j_sequence_start', 'cdr3_start', 'cdr3_end'.

**Labeled data subsets.** When running clustering experiments we tested various different subsets of the data, and used differing features as inputs for clustering. We select whether to use:
- chains: TRB, TRA or paired-chain TCRs
- feature: CDR3, CDR2+3, or CDR1+2+3.

We then subsequently only keep TCRs that bind antigens with at least 3 TCRs in the dataset, keep TCRs with length of less than 28 residues (CDR3,CDR3+2) or 35 (CDR3+2+1), and then removed duplicate TCRs based on the feature and chains that were to be used in an ML model.

**Unsupervised training data sets.** We train models in two regimes: in the first case we use a small fraction of the TRB TCR data (hereon called smallTRB) to allow us to sweep hyperparameters in a reasonable timeframe and for the final TRA and TRB models we use all of the TRA and TRB data respectively (largeTRA, largeTRB) for training the models. In both cases we remove duplicated sequences and split the data into train, validation and test sets with sizes 80%,10%,10% respectively. Exact number of unique sequences: largeTRB (94519890), largeTRA (5176669), smallTRB (4253395).

## TCR sequence formats and physicochemical projection

TCR sequences were represented as a combination of CDR regions (CDR3, CDR3+2, or CDR3+2+1). The CDR regions were joined with gap characters, and this was applied to either TRA-alone, TRB-alone to generate a feature for that TCR chain, and for paired-chain format the features for each chain were joined with a gap character.

To embed amino acid sequences of the TCR sequence, represented by the CDR combination being used, into physicochemical space we project using a normalized version of the physicochemical properties for each amino acid as they appear in Table 1 of ref. 64. Each amino acid is converted into a vector of 8 values, the seven physicochemical properties and a reserved feature indicating if the amino acid is a gap character, and normalize the values using z-score for each feature over the 20 amino acids. We can denote this space as $\{\boldsymbol{f}^a | a \in \text{amino acids}\}$.

After each amino acid in a sequence is converted a sequence is then represented by a 2D array with the length of the original amino acid sequence and width 8. We pad the end of the amino acid sequence with gaps such that all input data have the same array size, maximal size included is indicated in the following section and differs depending on which CDRs are included. This physicochemical encoding for a given sequence can be written as $x$ where $x_{i,j} = f_j^{a_i}$ where $a_i$ is the amino acid of the $i^{\text{th}}$ amino acid in the sequence.

We find that this physicochemical representation of TCRs is at baseline, before training of TCR-VALID, as capable as one-hot encoding at capturing relevant features for TCR-antigen clustering for all reference data sets when both are reduced to 16 PCA features and then clustered using DBSCAN (Supplementary Fig. 7). In fact we find that for high quality filtered TCR-antigen datasets such as our tcrvalid internal reference and the GLIPH2 filtered reference (VDJDB quality metric of 1 or above) that physicochemical representations outperform one hot encoding for TCR clustering.

A physicochemical representation of an amino acid sequence can be uniquely projected back to the original sequence, but with any minor alteration to physicochemical property at a given position the amino acid at the position can no longer be uniquely identified and must instead be a probability distribution over the amino acids. This probability assignment is made via a distance metric $\delta(\cdot, \cdot)$ on each physicochemical feature vector and the features of the true amino acids, subsequently normalized to a probability distribution. Namely, for a physicochemical representation $\tilde{x}$, $\tilde{x}_{i,j} = f^i_j$, the probability for each amino acid for position $i$ is $P(a_i = a | \boldsymbol{f}^i) = \delta(\boldsymbol{f}^i \boldsymbol{f}^a) / \sum_a \delta(\boldsymbol{f}^i \boldsymbol{f}^a)$. For the distance metric we used $\delta(\boldsymbol{f}^i \boldsymbol{f}^a) = 1 / (\|\boldsymbol{f}^i - \boldsymbol{f}^a\|_1 + \varepsilon)$ with $\varepsilon = 1e-6$. This allows any physicochemical feature array to be converted to a PWM over amino acids.

## Capacity-controlled VAE models

**Architecture.** TCR sequences projected into physicochemical arrays, $x$, are fed into an encoder model $q_{\theta_\mu, \theta_\sigma}(z | X) = \mathcal{N}(z | \mu(X; \theta_\mu), \sigma^2(X; \theta_\sigma))$ to generate latent space samples $z$, which are subsequently decoded by $p_\phi(X | z)$.

Since our input sequences are fairly short ($\leq 28$) we utilize a lightweight 3 layer CNN for our encoder with He normalization and stride of 1 for all 1D convolutions, with 32,64,128 channels and kernel widths 5,3,3. All convolutional layers are followed by Batch Normalization[65] and leaky ReLU activation. Following the the convolution the output is flattened to a vector and two single feed forward layers with output dimension 16 are used to construct the mean and log variance for the sampling of the latent representation to be passed to the decoder. For the decoder a ReLU activated feed forward layer constructs a $28 \times 128$ array on which 3 1D deconvolution layers are applied with channels 128,64,32 and kernel widths 3,3,5, and each is followed by Batch Normalization[65] and leaky ReLU. Final reconstructed physicochemical representation is generated by an 8 channel deconvolution layer with kernel width of 1 without activation.

**Training regime.** The loss function is that described by Burgess et al.[33]: $\mathcal{L} = \mathcal{L}_{\text{recon}} + \beta | \mathcal{L}_{\text{KL}} - n_L \mathcal{C} |$, where $n_L$ is the number of latent dimensions used and the loss terms used for sample $x_i$ were $\mathcal{L}_{\text{recon}} = \mathbb{E}_{q(z | x_i)} [\text{MSE}(x_i - p_\phi(X | z))]$ and $\mathcal{L}_{\text{KL}} = D_{\text{KL}}(\mathcal{N}(\mu(x_i; \theta_\mu), \sigma^2(x_i; \theta_\sigma)) || \mathcal{N}(0,1))$. In contrast to Burgess et al.[33] we do not adjust the capacity $\mathcal{C}$ during training, thereby fixing an average number of nats of information that a dimension of the latent space should aim to encode. We use a value of $\beta$ sufficient to enforce the average capacity is close to the objective, $\beta = 1$ was sufficient for $\mathcal{C} \geq 2$ whereas $\beta = 10$ was required for $\mathcal{C} = 1$. We note that since MSE loss is used for the reconstruction loss $\beta = 1$ does not carry the same meaning in terms of the ELBO that it does in the context of a Bernoulli output for binary X, see e.g., ref. 35.

We minimize the loss using the Adam optimizer[66] with a learning rate of 1e−3. For the smallTRB and largeTRA models we use early stopping on the validation data split with a patience of 10 epochs and restore the weights to the epoch at minimal validation loss. For the largeTRB model due to the size of data we used checkpointing at 10 epoch intervals and used the checkpointed model at approximate minima of the validation loss.

Training, validation and test datasets were saved in parquet format in either raw sequence format, ingested using HuggingFace's[67] read_parquet method and converted to physicochemical properties on the fly (for small TRB and largeTRA), or (for largeTRB) TRB sequences were first converted to physicochemical arrays in and saved in parquet format and then ingested using the read_parquet method.

## Metrics

**Clustering metrics.** A 'pure' cluster is one in which the modal antigen label of TCRs is the label of >90% of TCRs in the cluster, following the definition in ref. 43. If we consider such TCR clones to be 'clustering true positives', c-TP, while TCRs in clusters which don't fit this definition as 'clustering false positives', c-FP, and equivalently TCRs that aren't clustered at all to be 'clustering false negatives', c-FN, we can make the following analogies between clustering and classification metrics:

**c-Precision.** Of all clustered TCR clones, the percentage of TCR clones that are clustered into clusters that are 'pure'. Which can be written:

$$\text{c-Precision} = \frac{\text{c-TP}}{\text{c-TP} + \text{c-FP}} \qquad (1)$$

**c-CSI:** The Critical Success Index, of all TCRs how many were placed into 'pure' clusters:

$$\text{c-CSI} = \frac{\text{c-TP}}{\text{c-TP} + \text{c-FP} + \text{c-FN}} \qquad (2)$$

**Minimal cluster sizes.** In order to score the ability of a given TCR distance-metric in combination with a given clustering algorithm to accurately cluster TCRs, we take an approach similar to that of ref. 44 smaller clusters can skew the model performance metrics. We therefore chose to only consider clusters with at least 3 TCRs present. However, we also consider (Supplementary Fig. 2) the case where TCR clusters of only 2 TCRs are allowed and find that physicochemical properties alone perform as well as ismart and tcrdist on TRB chains with CDR123 considered.

**Continuity metrics.** We project two TCR sequences with identical CDR3s but differing CDR2 into physicochemical space $(x_1, x_2)$ and then into a models latent space $(z_1, z_2)$. We then linearly interpolate between $z_1$ and $z_2$ via : $z_i = \frac{i}{N-1} z_1 + \frac{N-1-i}{N-1} z_2$ for $i = 0, 1, \ldots, N-1$. Interpolated latent space representations $z_i$ are then decoded to physicochemical property maps $\tilde{x}_i$.

Inspired by Berthelot et al.[29] we devise measures of how far interpolated TCRs differ from the true manifold and expected pathway of TCRs between our endpoints. We identify if the CDR2 region of the interpolated TCR remains close to the true manifold of possible TRBV CDR2s and whether the CDR3 region differs significantly from the true CDR3 of both $\text{TCR}_1$ and $\text{TCR}_2$.

The distance of an interpolated TCR physicochemical representation to a true TRBV CDR2 can be written:

$$C_{i,q} = d(q, \tilde{x}_{i,CDR2}) \qquad (3)$$

$$q^*_i = \arg \min_q C_{i,q} \qquad (4)$$

$$D_{i,cdr2} = C_{i,q^*_i} \qquad (5)$$

where q is the physicochemical representation of the $q$th true CDR2 and $d$ is a distance metric in the physicochemical space and $\tilde{x}_{i,CDR2}$ is the CDR2 region of the physicochemical representation $\tilde{x}_i$. For $d$ we use $d(x, y) = \sum_j \|\boldsymbol{f}^{x,j} - \boldsymbol{f}^{y,j}\|_1$. $D_{i,cdr2}$ is the distance from an interpolated TCR CDR2 to the nearest true CDR2. We can average this quantity over a trajectory to get a score the the trajectory (smaller is better):

$$\overline{D}_{cdr2} = \frac{1}{N} \sum_{i=0}^{N-1} D_{i,cdr2} \qquad (6)$$

We can also assess how far the CDR3 changed from the true CDR3 at each point on the trajectory:

$$D_{i,cdr3} = d\left(x_{0,CDR3}, \tilde{x}_{i,CDR3}\right) \qquad (7)$$

$$\overline{D}_{cdr3},\text{raw} = \frac{1}{N} \sum_{i=0}^{N-1} D_{i,cdr3} \qquad (8)$$

$$\overline{D}_{cdr3} = \frac{1}{N} \frac{2}{D_{i,cdr3} + D_{N-1,cdr3}} \sum_{i=0}^{N-1} D_{i,cdr3} \qquad (9)$$

where $x_{0,CDR3}$ is the physicochemical representation of the CDR3 of the original TCR sequence. $\overline{D}_{cdr3}$ is the mean distance along the trajectory, normalized to the error in reconstruction at the start and end point of the trajectory. This is required to normalize across models with different reconstruction accuracy since we wish to assess changes in the manifold matching, rather than measure reconstruction accuracy which varies as capacity of the model is varied.

To compare models we generate many TCR interpolations and score them via $\overline{D}_{cdr2}$ and $\overline{D}_{cdr3}$. We generate TCR pairs with the same CDR2 but different CDR3, and in particular for each model we generated 310 interpolations - we chose 31 random CDR3s selected from the test set, and for each CDR3 interpolated between 10 random pairs of CDR2s of length 6 amino acids.

**Disentanglement Metric.** We train random forest classifiers (sklearn[68]) with TCR latent space representations as features and V or J gene as labels. In order to create the mean insert physicochemical value we average the physicochemical properties of all the amino acids attributed to the insert region as defined in the data preparation section. This value is then used to train a random forest regressor. In order to evaluate our classifiers we score them using weighted one versus 'rest' AUC ROC with stratified 5 fold cross validation. Hyperparameters for the random forest are chosen based on the cross-validation and RFs are retrained with those parameters on the full dataset. For training data for the RFs we use a random selection of 100k TCRs. We then take the random forest feature importances and apply the disentanglement metric as in Eastwood and Willliams[39].

**Uncertainty aware classification**
We used TCR-VALID models to generate representations for the TRB and TRA chains of paired chain TCRs in our labeled dataset restricted to TCRs with binding to antigens for which at least 100 unique TCRs were present in the dataset (8 antigens), and for 100k random unique chains from each of the unlabeled TRA and TRB datasets. The labeled data were further split into HLA-A*02 binding TCRs and non-HLA-A*02 binding TCRs. We built a feed-forward neural network with the following layers: dropout (25% retention), dense (128, ReLU), dropout (40% retention), dense (128, ReLU), dropout (40% retention), dense (8, softmax). We constructed input batches of 32 labeled TCRs and 512 randomly selected unlabeled TCRs and trained the neural network with loss function[56]:

$$\mathcal{L} = \mathbb{E}_{\mathcal{P}_{in}(\hat{x},\hat{y})} \left[ -\sum_{c} \mathcal{I}_{\hat{y},c} \omega_c \log(P_\theta(y=c|\hat{x})) \right]$$
$$+ \alpha \mathbb{E}_{\mathcal{P}_{out}(\hat{x})} \left[ \sum_{c} \text{KL}\left(U || \log(P_\theta(y=c|\hat{x}))\right) \right] \qquad (10)$$

Where we apply weights, $\omega_c$, to the cross-entropy loss to combat dataset imbalance, $\mathcal{I}_{\hat{y},c}$ is the indicator function, and $\alpha$ is a tunable hyperparameter. $\omega_c$ are calculated using sklearǹs 'compute_class_weight' function. We only trained the model on the HLA-A*02

associated TCRs, and used the non-HLA-A*02 TCRs as an OOD test set. During training we apply an early stopping criteria with a patience of 5 epochs on the validation set. 75% of the in-distribution data was used for training, and 12.5% for validation and testing respectively.

AUROC for in-distribution peptide classification are calculated in a one vs rest fashion per peptide on the left out test set. As a measure for detecting OOD we data we calculate the classifier confidence as $N \max\{P_\theta\}/(N-1)$, where $P_\theta$ are the multiclass probabilities for a sample and N is the number of classes. We then use these confidence scores as effective probabilities for measuring whether data is ID or OOD and use these probabilities to construct an ROC curve and calculate the AUROC for OOD detection. Reported AUROC are calculated as the mean over 10 Monte Carlo cross-validation splits. p-values and U statistics between AUROC of different methods are calculated over the Monte-Carlo splits via a two-sided Mann-Whitney U test, and the (common-language) effect size between two groups of 10 values can be calculated from the quoted U statistics by $U/100$.

**Comparator methods**
**Physicochemical properties.** We project the residues of TCRs into their physicochemical properties, constructing a 2D image for each TCR as we do for VAE input, and then flatten these 2D images to 1D vectors. These 1D representations are constructed for the subset of labeled TCR data of interest, and clustering is then performed on these representations via Euclidean distance via DBSCAN[46] via scikit-learn's implementation[68] (similarly for all DBSCAN implementations discussed below).

**PCA on physicochemical properties.** These 1D physicochemical representations are constructed for the subset of labeled TCR data of interest, and for a subset of the unlabeled TCRs. We fit PCA to the unlabeled TCR representations, and project antigen-labeled TCRs into the PCA dimensions. TCR clustering is then performed on PCA representations with DBSCAN. We use inverse PCA transform to project TCR representations in PCA space back into physicochemical space, and subsequently convert from physicochemical space to probability distribution over residues to construct generated TCRs logos, via weblogo[69], from any position in the PCA latent space.

**PCA on one-hot encoding.** We one-hot encode the TCR sequence residues of the same format as those projected into physicochemical space with a gap character and concatenate them into a 1D array. We then fit PCA to the unlabeled one-hot encoded TCR sequence representations into the same number of dimensions as TCR-VALID latent dimensions (and the physicochemical properties PCA) before undertaking the same TCR clustering procedures using DBSCAN.

**tcr-dist.** We used tcr-dist[47] (https://github.com/phbradley/tcr-dist, SHA: 1f5509a, license: MIT) to extract the tcr-dist distances between all pairs of clones in each set of input data with a minor modification to allow the computation to be parallelized while returning the same TCR-TCR distance matrix (all code shared, see Code Availability). We then used the TCR-TCR distance matrices saved by tcr-dist to employ our own clustering on the resulting graph, in this paper we used DBSCAN and we performed such clustering for TRB-only, TRA-only and paired-chain clustering. We found this to be faster than the original tcr-dist, and DBSCAN has previously been found to be the best clustering algorithm on the tcr-dist graph by others[44]. This was essential for comparison across many thresholds and datasets including spike-in. For timing we used the more recent and significantly faster tcrdist3[49], with post-hoc DBSCAN for timing of clustering.

**iSMART.** We use iSMART[45] (https://github.com/s175573/iSMART, SHA: 1f2cbd2, license: GPL3.0) to cluster TRB-only data, as it was not designed to cluster TRA or paired-chain TCR data. We tune the internal

distance parameter, `threshold`, within their clustering method to tune the size of the clusters.

**GLIPH2.** We use the GLIPH2 webtool[43] (http://50.255.35.37:8080) to cluster TRB-only data. As the webtool was not designed to cluster TRA or paired-chain TCR data. We process the reference datasets and spike in datasets to fit the webtools format and set subject to NA and count to 1. The output of the GLIPH2 webtool clustering is then processed to account for the multiple assignment that the tool allows, that is TCR clones can be assigned to more than one cluster which makes fair comparison with other tools challenging. Namely, for TCRs assigned to more than one cluster we only kept the occurrence of each TCR from the largest cluster in which it is found and removed the other duplicate occurrences. The newly formed singleton clusters from the duplicate removal were then re-assigned as unclustered. A comparison of the clustering performance with and without this correction can be found in Supplementary Figs. 2–6.

**DeepTCR-VAE.** In order to use deepTCR[13] to cluster TCRs (TRB and TRA chains), we train and embed a deepTCR VAE using the TCR-antigen reference dataset as described in ref. 13 for clustering TCRs. We then used the representations to cluster TCR-antigen pairs using DBSCAN for minimum cluster size of 3. We found this tool required large amounts of RAM and so limited our analysis to clustering without further spike-in data.

**clusTCR.** We use clusTCR[44] to cluster TRB-only data, as it was not designed to cluster TRA or paired-chain TCR data. The output of the clustcr clustering is then processed like GLIPH2 clustering output to account for the multiple assignment that the tool allows.

**TCR-BERT.** We use TCR-BERT[10] in two modes, their internal clustering method and DBSCAN clustering applied to a distance graph defined by distance between the TCR-BERT embeddings. TCR-BERT embeddings are calculated from the output of the eighth layer of the transformer model, as used within the original publication due to this layer's outputs being optimal[10]. We use the script `embed_and_cluster.py` to perform tcr-bert clustering, tuning the Leiden resolution to adjust the cluster sizes. It is worth noting that this clustering assigns a cluster to all TCRs, in contrast to DBSCAN which allows TCRs to not belong to any cluster. We also collected the TCR-BERT embeddings from the output of the eighth layer (using 'mean' aggregation as per the tcr-bert default) and used Euclidean distance between these embedding representations to build a distance graph on which we could apply DBSCAN clustering.

**ESM.** We used ESM-1b[26] to generate representations of TCR sequences in several ways. We gave the TCR sequences as input with either the CDR3 alone, or the CDR2 and CDR3 joined with a single gap. To generate TCR distances from the TCR representations generated by ESM1b we employ Euclidean distances between the representations, either on the entire representation (flattened to 1D) or the mean-pooled representation. DBSCAN is used as the clustering algorithm on this TCR distance metric.

**DeepTCR-classification.** DeepTCR was used in its classifier mode as described in ref. 13, trained only on the HLA-A*02 associated TCRs. Model performance on in-distribution TCRs, and OOD (non-HLA-A*02 associated TCRs) was evaluated using Monte Carlo 10 fold cross validation. 75% of the in-distribution data was used for training, and 12.5% for validation and testing respectively. Model confidence was calculated in the same manner as for TCR-VALID.

## Spike in TCR-clustering

As in ref. 43, in order to test the robustness of out TCR clustering approaches we spiked in irrelevant TCRs at folds 1-5x the reference

data set size. In order to truly account for the spike in TCRs as irrelevant we labeled them with unique hashed values as their peptide label. In this way they cannot form pure clusters of their own and are counted against the purity of any cluster of labeled antigen.

## Timing analysis

We benchmarked timing of various TCR clustering approaches we used the python package time. Performance was evaluated on a 16-core CPU for all tools. For tcrdist we used the faster, numba optimized, implementation of tcr-dist (tcr-dist3). For TCR-VALID we separately timed the embedding, clustering via DBSCAN, and distance matrix calculation, for tcr-dist3 we also additionally calculated the distance matrix timing without clustering. Timing was calculated as the mean over 5 repeats for fewer than 10,000 TCRs (or 400 TCRs for iSMART and tcr-dist3) otherwise as a single repeat for larger numbers of TCRs.

## Reporting summary

Further information on research design is available in the Nature Portfolio Reporting Summary linked to this article.

## Data availability

Collected repertoire level TCR data from the iReceptor Gateway[59] (https://gateway.ireceptor.org/login) and VDJServer[60] (https://www.vdjserver.org) and is publicly available. A list of the repertoire_id`s of the repertoires used in this study are included in our github (https://github.com/peterghawkins-regn/tcrvalid). Collected paired-chain TCRs with known cognate antigens from two sources; those associated with[8] (https://doi.org/10.1126/sciadv.abf5835, https://github.com/regeneron-mpds/TCRAI/tree/main/data) and VDJdb[61,62] (all human paired-chain TCRs with a quality 'score' of at least 1 (accessed October 2021) https://vdjdb.cdr3.net) are also publicly available, the dataset has been deposited in our Github repository. We additionally collected TCR-antigen reference data, and CD4 spike in TCRs, from the supplementary material of Huang et al.[43] (10.1038/s41587-020-0505-4), the dataset and spike-in splits used are available in our Github repository. Source data are provided with this paper.

## Code availability

The tcrvalid package, and pre-trained models, are available at https://github.com/peterghawkins-regn/tcrvalid under Apache-2.0 license.

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

## Author contributions

A.Y.L. and P.G.H. conceived the study. P.G.H. and G.S.A. supervised the study. A.Y.L. and P.G.H. designed and performed experiments and data analysis. D.Sc and P.G.H. performed data analysis of transformer methods. A.Y.L., D.Sc, N.T.G., J.C.W., D.Sk, G.S.A. and P.G.H. contributed to discussions of results and methods. A.Y.L., D.Sc, N.T.G., P.G.H. wrote the manuscript with input from J.C.W., D.Sk and G.S.A.

## Competing interests

The authors are employees of, and have stock and stock options in, Regeneron Pharmaceuticals Inc. G.S.A. is an officer of Regeneron Pharmaceuticals Inc.
