## [Peer Review File · Nature Communications]

Designing meaningful continuous representations of T cell receptor sequences with deep generative modelsREVIEWER COMMENTS

Reviewer #1 (Remarks to the Author):

This is an interesting paper describing an AI model termed “Capacity controlled C β -Variational Auto Encoder (VAE) or C β -VAE in short. The authors developed and configured this C β -VAE model to transform the physicochemical features of CDR2-CDR3 amino acid sequences into 16-dimension (16D) latent representations. This 16D representations allows them to do de novo TCR sequence generation. Through different matrices, the author believed that the 16D representations of CDR2-CDR3 sequences are low-dimensional, continuous, disentangled and sufficient informative. To demonstrate the usage of 16D representations, the authors used DBSCAN (a clustering algorithm) to generate clusters of TCRs on top of the 16D representations. To benchmark their model, the author compared their results with iSMART, and tcr-dist two sequence-based methods, TCR-BERT and ESM-1b two AI-based methods.

Overall, it is a very innovative approach. However, I am not sure how good the 16D representations for CDR2-CDR3 sequences are. Figure 4 show the comparison of results for TCR-VALID, iSMART and tcr-dist, which is not very impressive since the difference between TCR-VALID and iSMART is quite trivial. In a published report (Huang et al. Nat. Biotech 2020) iSMART gave very poor results, especially compared to GLIPH2, one of the most commonly used (and published) TCR clustering methods. At a minimum the authors should compare their method with GLIPH2, and with different levels of noise spiking-in, a key test of an algorithm is how it deals with noisy data.

A minor point, line 344 says that a CDR3 ends with the amino acid F. This claim is wrong. CDR3s coded by TRBJ2-7*02 end with the amino acid V. CDR3s coded by TRAJ33*01 end with the amino acid W, coded by TRAJ35*01 end with the amino acid C.

Reviewer #2 (Remarks to the Author):

The authors propose an interesting encoding of TCRs beta chains to real vectors, with a few neat tricks, such as the representation of V genes by their CDR2 region unifying the

representation of CDR3 and V (instead of a one hot used in most previous publications). They also a C-beta AE, which may be smarter than many of the existing models that use VAE or similar models.

The authors also provide some interesting tests, and the capacity to generate new TCRs. The manuscript is well written and very easy to understand. This paper is TCRdist, DeepTCR, GIANA, iSMART, GLIPH, ELATE, as well as the Pan-Peptide Meta Learning of Gao et al recently in Nat. ML, and all the language models recently developed such as TCR-Bert, using the roberta framework. All these models and a few mores that I may have missed produce similar encoding, many with much more extensive testing than the one presented here. While some comparison with previous models is performed (e.g. idist and tcr-bert), the comparison is very limited, not presented in the main text, and does not contain most of the standard tests proposed by other models.

Beyond that, practically all the TCR-peptide prediction algorithms contain an embedding of the TCR, and as such are similar work. Especially pan-peptide binders (some of which are cited here, but others are not). Thus, at least for the peptide binding profiles, those could be compared.

In short, this is a really interesting paper, but it is not convincing at all that it adds anything significant to the current results. It may actually well be the case, but this is neither shown nor explained by a proper comparison to existing tools, and given the resemblance of the current model to many existing ones, it is not clear if there is any novelty.

Reviewer #3 (Remarks to the Author):

The authors propose a capacity constrained autoencoder to project TCR sequences into a low-dimensional latent space.

1. The authors need to clearly articulate what their method does better than other methods and provide statistically validated evidence of these improvements
2. As far as I can tell there are no statistical tests in this paper. While general statements are useful, specific benchmarks with other methods need to be performed to help the reader understand the superior aspects of the present approach.

3. Certain of the choices the authors made such as the representation of amino acids as 7 physicochemical features is not compared to the alternative of one-hot encodings. It would be helpful to the reader to understand the marginal benefit of such choices.
4. Please provide a biologically relevant benchmark for latent space smoothness and show that your method is better than DeepTCR or other methods.
5. Once again, please provide a statistical test of your claim of superior clustering, and compare with another autoencoder baseline such as DeepTCR.

Thank you.

REVIEWER COMMENTS

Reviewer #1 (Remarks to the Author):

This is an interesting paper describing an AI model termed “Capacity controlled C β -Variational Auto Encoder (VAE) or C β -VAE in short. The authors developed and configured this C β -VAE model to transform the physicochemical features of CDR2-CDR3 amino acid sequences into 16-dimensional (16D) latent representations. This 16D representation allows them to do de novo TCR sequence generation. Through different matrices, the author believed that the 16D representations of CDR2-CDR3 sequences are low-dimensional, continuous, disentangled and sufficiently informative. To demonstrate the usage of 16D representations, the authors used DBSCAN (a clustering algorithm) to generate clusters of TCRs on top of the 16D representations. To benchmark their model, the author compared their results with iSMART, and tcr-dist two sequence-based methods, TCR-BERT and ESM-1b two AI-based methods.

Overall, it is a very innovative approach.

We thank the reviewer for their thoughtful consideration of our work and are glad that they found the approach innovative.

Before addressing the concerns below, we would like to address the reviewer’s note that we use various metrics to quantify the smoothness and disentanglement of this low-dimensional space. We have slightly adjusted the manuscript to further highlight the importance of this element of our work. Namely, while other autoencoding and VAE approaches have been developed for TCRs, none had to our knowledge explored the extent which the dimensionality and information in the latent space could be limited while retaining desirable properties of a latent space. The in-depth investigation of these properties, together with benchmarking of downstream tasks of clustering and classification, contributes to the overall value and innovation of TCR-VALID in comparison with previous approaches.

We discuss updates to TCR clustering within the manuscript below, and here we wanted to note we have also extended our analysis of TCR classification and out-of-distribution (OOD) detection, including a comparison to the deepTCR classifier. Notably, the deepTCR classifier uses a much larger internal representation than the TCR-VALID model and is not informationally constrained. TCR-VALID performs very similarly in classification despite these constraints. We further show that it is possible to improve OOD detection using human repertoire TCRs, using a true OOD set by splitting TCRs into those associated with HLA-A*02 or not.

However, I am not sure how good the 16D representations for CDR2-CDR3 sequences are. Figure 4 shows the comparison of results for TCR-VALID, iSMART and tcr-dist, which is not very impressive since the difference between TCR-VALID and iSMART is quite trivial. In a published report (Huang et al. Nat. Biotech 2020) iSMART gave very poor results, especially compared to GLIPH2, one of the most commonly used (and published) TCR clustering methods. At a

minimum the authors should compare their method with GLIPH2, and with different levels of noise spiking-in, a key test of an algorithm is how it deals with noisy data.

We thank the reviewer for suggesting we undertake a spike-in analysis. We have conducted a spike-in analysis mirroring that of GLIPH2 and extended our benchmarking to include the GLIPH2 tool in addition to other tools: clusTCR and deepTCR (Fig 4). We have performed spike-in analysis using the background CD4 TCRs used in the GLIPH2 analysis together with both the labeled dataset used in our manuscript, and the labeled GLIPH2 dataset (Supp Figs. 3-6). We additionally performed clustering on a subset of the GLIPH2 dataset limited to those TCR-Ag relationships with quality score of 1 or more according to VDJdb (the original source of the dataset). We find that all clustering tools perform worse on the GLIPH2 dataset, and once limiting more stringently on quality the performance across methods is closer to that in our dataset. This interestingly suggests that continued measurement and assessment of high quality TCR-antigen associations are of great importance for the community in developing clustering methods.

One important aspect of our clustering, and spike-in, analysis lies in the way we assess the performance of the clustering tools. We do not use tools only using the out-of-the-box parameters (unless we were unable to adjust the key distance threshold for some reason, as explained in methods for GLIPH2 and clusTCR). This is very important to properly benchmark tools against one another and drove our selection of this method of comparison in the originally submitted manuscript. In particular, consider comparing two classifiers with output prediction on [0,1]. One might typically assess the AUROC rather than the specificity or sensitivity alone, since assessing only the specificity at one threshold for each model would not allow one to thoroughly compare the models' relative performance.

In the case of the comparison between iSMART, tcr-dist, and GLIPH2 in the supplementary material of the GLIPH2 paper: it is shown that GLIPH2 clusters a higher percentage (~19% vs ~9%) of unique CDR3s into clusters of high purity than iSMART with the default distance threshold choice. At this threshold, it is true that iSMART has lower CSI, but it also has higher precision. If we track both precision and CSI as one tunes the distance threshold and compare to the precision of GLIPH2 we see that GLIPH2 performance lies close to the locus of iSMART performance. That is, at a different threshold (7.0 instead of default 7.5) iSMART performance is more similar to GLIPH2 in terms of both CSI (19.3% (GLIPH2) vs 17.5%) (as measured by the GLIPH2 authors) and precision (70.9% vs 80.4%). As noted above, for GLIPH2 we were unable to scan the effective distance threshold, as discussed in methods.

For GLIPH2 (as well as the original tcr-dist clustering algorithm and clusTCR) TCR clones can be clustered into multiple clusters. This makes comparison of the models challenging. As described in the methods, we evaluate clustering with clones only retained in the largest cluster to which they belong in these cases. For GLIPH2 we include performance with and without this correction (Supp.Figs2-6) for the benefit of the reader who may want to understand performance in the two cases. Notably, when we apply this correction to GLIPH2 when clustering the GLIPH2 reference dataset the CSI, precision fall further to (15.6%, 66.3%).

In addition, and of equal importance, to the clustering performance, we have profiled the runtime of clustering (and distance matrix) calculation. The fastest method we profiled was clusTCR which uses the FAISS tool to perform fast neighbor search using approximate nearest neighbor (ANN) algorithm(s). However, clusTCR's clustering performance is based on Hamming Distance, and performs poorly in our benchmarking. We find that TCRVALID is much faster than iSMART and the latest, optimized tcrdist3 algorithm. The majority of clustering time for TCRVALID occurs in the DBSCAN clustering step, not in the embedding of the TCRs, and is driven by the distance calculations (Supp Fig 8). It is therefore possible that the use of ANN algorithms would provide a speed boost for TCRVALID as in the case of clusTCR.

In summary for TCR clustering, we included a spike-in analysis, extending the clustering tools benchmarked against, and profiled the time complexity these tools. We have shown that TCR-VALID performs clustering similarly to state-of-the-art tools on high quality TCR datasets. The TCR-VALID clustering is faster than distance-based tools tcr-dist3 and iSMART, and while slower than clusTCR the clustering performance is better. Our method for evaluating TCR clustering takes into account the importance of distance thresholds and both precision and CSI which is essential for a thorough comparison. The TCR-VALID clustering occurs in a low dimensional space that we carefully constructed, and thoroughly profiled, and as such allows for interrogation and de novo generation of TCRs with or without clustering.

A minor point, line 344 says that a CDR3 ends with the amino acid F. This claim is wrong. CDR3s coded by TRBJ2-7*02 end with the amino acid V. CDR3s coded by TRAJ33*01 end with the amino acid W, coded by TRAJ35*01 end with the amino acid C.

We thank the reviewer for this important note, we have changed the text in the manuscript accordingly.

Reviewer #2 (Remarks to the Author):

The authors propose an interesting encoding of TCRs beta chains to real vectors, with a few neat tricks, such as the representation of V genes by their CDR2 region unifying the representation of CDR3 and V (instead of a one-hot used in most previous publications). They also use a C-beta AE, which may be smarter than many of the existing models that use VAE or similar models. The authors also provide some interesting tests, and the capacity to generate new TCRs. The manuscript is well written and very easy to understand.

We thank the reviewer for their careful consideration of our manuscript, and we are glad to hear they found it well-written and found that we had interesting methodologies and tests.

This paper is TCRdist, DeepTCR, GIANA, iSMART, GLIPH, ELATE, as well as the Pan-Peptide Meta Learning of Gao et al recently in Nat. ML, and all the language models recently developed such as TCR-Bert, using the roberta framework. All these models and a few more that I may have missed produce similar encoding, many with much more extensive testing than the one presented here.

While some comparison with previous models is performed (e.g. idist and tcr-bert), the comparison is very limited, not presented in the main text, and does not contain most of the standard tests proposed by other models.

We thank the reviewer for suggesting broader benchmarking of our method. We have extended our comparisons with other methods, namely including DeepTCR, GLIPH2 and clusTCR, as well as those originally included: iSMART, tcr-dist, TCR-BERT, and ESM, the latter two of which are trained in BERT/ROBERTa framework. We were unable to include GIANA due to licensing issues. We have additionally extended our comparison to include a spike-in analysis of irrelevant TCRs to monitor the performance drop as these TCRs are spiked in. We analyzed the effect of spike-in performance degradation using two labeled TCR datasets (Fig.4, Supp.Figs.2-6).

One important aspect of our clustering, and spike-in, analysis lies in the way we assess the performance of the clustering tools. We do not use tools only using the out-of-the-box parameters (unless we were unable to adjust the key distance threshold for some reason, as explained in methods for GLIPH2 and clusTCR). This is very important to properly benchmark tools against one another and drove our selection of this method of comparison in the originally submitted manuscript. In particular, consider comparing two classifiers with output prediction on $[0,1]$ where one might typically assess the AUROC rather than the specificity or sensitivity alone since assessing only the specificity at one threshold for each model would not allow one to thoroughly compare the models' relative performance.

For TCR clustering an increasingly common way to assess the performance is to assess the purity of clusters and track the number of TCRs placed into pure clusters (as analyzed by GLIPH2, and clusTCR authors, and with similar analyses in GIANA and in the supplement of DeepTCR). Importantly we track both the CSI (the fraction of TCRs placed in pure clusters) and

precision of the clustering models and do so across a range of the models' internal distance thresholds. This allows us to track that clustering performance and make a fair comparison of the locus of performance attainable with differing thresholds. Many tools perform similarly once we account for this, and TCR-VALID performs similarly to iSMART and tcr-dist while outperforming other methods. Given that different TCR clustering use cases might call for a greater emphasis precision or CSI we believe our benchmarking curves allow the reader to see in which regime each tool excels in to pick the most relevant one. We only make claims about TCR-VALID outperforming other tools when its precision vs CSI curve is consistently above that of another tool when we scan the relevant "radius" parameter, in this way we are not picking a particular set radius, CSI or precision to benchmark tools which can be misleading.

In addition to the clustering performance, we have profiled the runtime of clustering (and distance matrix) calculation. The fastest method we profiled was clusTCR which uses the FAISS tool to perform fast neighbor search using approximate nearest neighbor (ANN) algorithm(s). However, clusTCR's clustering performance is based on Hamming Distance, and performs poorly in our benchmarking. We find that TCRVALID is much faster than iSMART and the latest, optimized tcrdist3 algorithm. The majority of clustering time for TCRVALID occurs in the DBSCAN clustering step, not in the embedding of the TCRs, and is driven by the distance calculations (see Supp Fig 8). It is therefore possible that the use of ANN algorithms would provide a speed boost for TCRVALID as in the case of clusTCR.

Beyond that, practically all the TCR-peptide prediction algorithms contain an embedding of the TCR, and as such are similar work. Especially pan-peptide binders (some of which are cited here, but others are not). Thus, at least for the peptide binding profiles, those could be compared.

We thank the reviewer for highlighting an oversight in the referencing of recent pan-peptide binders. We have broadened our discussion of this in the introduction, and in the classification section, of our manuscript to highlight the importance of this research direction.

We are particularly interested in the case of applying a classifier to large repertoires of TCRs which is a very common use case (an extension of consideration of clustering to large datasets, including with spiked-in random TCRs). In this case a TCR classifier can be trained on some set of in-distribution (ID) TCRs, and a threshold may be chosen under the consideration of FPR, TPR. However, it is important to consider how the model behaves on TCRs that do not bind one of the antigens in the training set (OOD TCRs). When choosing a model threshold, it is therefore important to also consider whether you can exclude predictions based on a model's confidence, namely via OOD detection. While the important recent developments in predicting the cognate antigen of TCRs even for OOD antigens (and even TCRs) are no doubt important, one would still want to place some confidence bound on predictions over large datasets. We therefore decided to limit our study to understanding how the TCR-VALID latent space performs in ID classification and OOD detection, as we believe this remains an important use case.

Extending our comparisons for classification and OOD detection, we included a benchmarking against DeepTCR in its classification mode. This extends the comparison to deepTCR in its VAE

mode for clustering that we also performed. To assess ID performance and OOD detection we strictly separated TCRs into two groups (HLA-A*02 associated and not) and trained classifiers only on the HLA-A*02 portion. Comparing DeepTCR and TCR-VALID, we find very similar ID classification performance despite the more heavily constrained latent space. We further suggested and evaluated a method to improve OOD detection without affecting ID classification performance.

Regarding the embeddings themselves, we are not aware of any combined study of informational constraints, latent space smoothness, and disentanglement of other modelling approaches. DeepTCR does not constrain the capacity of the latent embeddings directly but does use a parameter to control the weighting between reconstruction and Kullback-Leibler loss. This weighting effectively controls the information in the latent space, though without a specific target. In DeepTCR the relative weight of the KL loss is set to $1e-3$, making it very close to an autoencoder. As we show in Figs 2-3 autoencoders perform poorly both in landscape smoothness and in disentanglement. As such, those models will generate TCRs without realistic germline components with small motion away from the training data, and the meaning behind the dimensions (e.g represent V gene, pseudo-random insert region, etc) are much more entangled. We provide a roadmap of how others may quantitatively assess these smoothness and disentanglement properties of the latent spaces of TCRs in other modelling strategies which may be particularly useful for generative modeling. It is possible that quantifying the disentanglement and smoothness in pan-peptide models may aid in interrogation of model predictions and any downstream generative applications from such models, e.g. generation of TCRs binding an antigen of interest for which no current cognate TCRs are known.

In short, this is a really interesting paper, but it is not convincing at all that it adds anything significant to the current results. It may actually well be the case, but this is neither shown nor explained by a proper comparison to esembla tools, and given the esemblance of the current model to many existing ones, it is not clear if there is any novelty.

We are glad the reviewer found our methods interesting and we believe our updates to the manuscript, as described above, further highlight the novelty and importance of this work. To briefly summarize some key aspects of our work:

For a generative model of TCRs to be useful for optimization with limited experimental resources, it is desirable that representations be low-dimensional and ideally smooth. BERT based encodings of TCRs are large dimensional and not inherently generational, and existing approaches of VAE methodologies have typically used large dimensional spaces and not thoroughly evaluated latent space smoothness. It is therefore important to understand whether it is possible to build a low dimensional representation of TCRs that is smooth and understand how to quantify that. We developed methods to quantify the smoothness of the latent space of TCR-VALID and carefully controlled the informational capacity of the model to tune these properties. We then confirmed that proximity in TCR-VALID space corresponds with function by performing TCR clustering. With our extended benchmarking and careful consideration of how to evaluate TCR-clustering we show that TCR-VALID is not limited by its low-dimensional and

informationally constrained space. Additionally, TCR-VALID is faster than other commonly used tools tcr-dist3 and iSMART, and outperforms the faster clusTCR in clustering performance. In addition, we show strong classification performance and OOD detection despite the informational and dimensional constraints. We therefore believe our manuscript is the first to show that a low-dimensional and capacity-constrained representation of TCRs can perform fast and accurate TCR-clustering and does so in a continuous space that allows for interrogating those clusters and for de novo TCR generation, which may further pave a way for sequence optimization due to the construction of the space.

Reviewer #3 (Remarks to the Author):

The authors propose a capacity constrained autoencoder to project TCR sequences into a low-dimensional latent space.

1. The authors need to clearly articulate what their method does better than other methods and provide statistically validated evidence of these improvements

Firstly, we would like to thank the reviewer for reading and evaluating the research presented in our manuscript. To briefly summarize some key aspects of our work:

For a generative model of TCRs to be useful for optimization with limited experimental resources, it is desirable that representations be low-dimensional and ideally smooth. BERT based encodings of TCRs are large dimensional and not inherently generational, and existing approaches of VAE methodologies have typically used large dimensional spaces and not thoroughly evaluated latent space smoothness. It is therefore important to understand whether it is possible to build a low dimensional representation of TCRs that is smooth and understand how to quantify that. We developed methods to quantify the smoothness of the latent space of TCR-VALID and carefully controlled the informational capacity of the model to tune these properties. We then confirmed that proximity in TCR-VALID space corresponds with function by performing TCR clustering. With our extended benchmarking and careful consideration of how to evaluate TCR-clustering we show that TCR-VALID is not limited by its low-dimensional and informationally constrained space. Additionally, TCR-VALID is faster than other commonly used tools tcr-dist3 and iSMART, and outperforms the faster clusTCR in clustering performance. In addition, we show strong classification performance and OOD detection despite the informational and dimensional constraints. We therefore believe our manuscript shows low-dimensional representations of TCRs can perform fast and accurate TCR-clustering and does so in a space that allows for interrogating those clusters and for de novo TCR generation, which may further pave a way for sequence optimization due to the construction of the space.

We will address the validated tests in our response to point 2.

2. As far as I can tell there are no statistical tests in this paper. While general statements are useful, specific benchmarks with other methods need to be performed to help the reader understand the superior aspects of the present approach.

We thank the reviewer for raising this point, and of course in general we strongly agree with this. In our study of TCR classification and OOD detection we have included a comparison with DeepTCR. In that instance we do provide statistical tests between performance of TCR-VALID and DeepTCR in addition to between parameter choice for balancing OOD-detection and ID-classification that is provided in TCR-VALID.

For TCR clustering the problem is more complex. For TCR clustering an increasingly common way to assess the performance is to assess the purity of clusters and track the number of TCRs placed into pure clusters (as analyzed by GLIPH2, and clusTCR authors, and with similar analyses in GIANA and in the supplement of DeepTCR). Importantly we track both the CSI (the fraction of TCRs placed in pure clusters) and precision of the clustering models and do so across a range of the models' internal distance thresholds. This allows us to track that clustering performance and make a fair comparison of the locus of performance attainable with differing thresholds. Many tools perform similarly once we account for this, and TCR-VALID performs similarly to iSMART and tcr-dist while outperforming other methods.

We believe that providing the full curves tracking the CSI and precision over the full range of thresholds (where possible, unfortunately we were unable to tune a distance for clusTCR and GLIPH2) is the fairest means of comparison. In part, this is because the choice of clustering tool will depend somewhat on the use-case: e.g. for a given requirement on precision a different tool may be selected that offers the highest CSI at that precision. We only make claims about TCR-VALID outperforming other tools when its precision vs CSI curve is consistently above that of another tool when we scan the relevant "radius" parameter, in this way we are not picking a particular set radius, CSI or precision to benchmark tools which can be misleading.

Additionally, speed constraints have to be taken into account for clustering large datasets, and this may preclude the most accurate clustering tools. In addition to the clustering performance, we have profiled the runtime of clustering (and distance matrix) calculation. The fastest method we profiled was clusTCR which uses the FAISS tool to perform fast neighbor search using approximate nearest neighbor (ANN) algorithm(s). However, clusTCR's clustering performance is based on Hamming Distance, and performs poorly in our benchmarking. We find that TCRVALID is much faster than iSMART and the latest, optimized tcrdist3 algorithm. The majority of clustering time for TCRVALID occurs in the DBSCAN clustering step, not in the embedding of the TCRs, and is driven by the distance calculations (Supp Fig 8). It is therefore possible that the use of ANN algorithms would provide a speed boost for TCRVALID as in the case of clusTCR.

We have shown that TCR-VALID performs clustering similarly to state-of-the-art tools on high quality TCR datasets. The TCR-VALID clustering is faster than distance-based tools tcr-dist3 and iSMART, and while slower than clusTCR the clustering performance is better. Our method for evaluating TCR-clustering takes into account the importance of distance thresholds and both precision and CSI which is essential for a thorough comparison.

3. Certain of the choices the authors made such as the representation of amino acids as 7 physicochemical features is not compared to the alternative of one-hot encodings. It would be helpful to the reader to understand the marginal benefit of such choices.

One-hot encodings of the sequence could also be used for the encoding of the TCRs, though one would still have a choice in how much information could be retained per latent dimension in the latent space. Since encoding TCRs by two dictionaries that have a one-to-one mapping

between amino acids and codes, we may not expect a large difference in the clustering performance. We have performed PCA to 16 dimensions based on a one-hot encoding and physicochemical encoding of TCRs, and then clustered TCRs and scored the performance. We find that the two encodings provide similar performance, though the physicochemical encoding does slightly outperform (precision vs CSI curve consistently above) the one-hot encoding on the high-quality datasets, as can be seen in Supplementary figure 7. It is possible this difference is driven by the extra information gained by the physicochemical embeddings (i.e. that charged residues such as R and K are more similar), or could be due to the additional dimensional reduction prior to the application of PCA.

4. Please provide a biologically relevant benchmark for latent space smoothness and show that your method is better than DeepTCR or other methods.

We are not aware of any combined study of informational constraints, latent space smoothness, and disentanglement of other modelling approaches. DeepTCR does not constrain the capacity of the latent embeddings directly but does use a parameter to control the weighting between reconstruction and Kullback-Leibler loss. This weighting effectively controls the information in the latent space, though without a specific target. In DeepTCR the relative weight of the KL loss is set to $1e-3$, making it very close to an autoencoder. As we show in Figs 2-3 autoencoders perform poorly both in landscape smoothness and in disentanglement. As such, those models will generate TCRs without realistic germline components with small motion away from the training data, and the meaning behind the dimensions (e.g. represent V gene, pseudo-random insert region, etc) are much more entangled.

Smoothness is evaluated by methods we developed to address this question, wherein we specifically applied biologically relevant knowledge to methods developed in the ML community to assess the smoothness of latent spaces. Namely, via interpolating linearly in latent space between TCRs with identical CDR3 but different CDR2. In this case, for a smooth space that could be used for TCR generation, one would hope that as one moves along such a trajectory: i) the CDR2 should never become too far from the CDR2s seen in human populations, and ii) that since the CDR3 is fixed at either end of the trajectory that the CDR3 should not change for generated TCRs along such a trajectory. We therefore quantified these expectations by measuring: i) the proximity of generated TCRs along the interpolation to the manifold of true human TCRs (i.e. does the generated TCR have a CDR2 close to a real CDR2, and ii) the distance of generated CDR3 from the expected CDR3 along the interpolation.

It is worth noting that DeepTCR, in addition to behaving like an autoencoder has a much larger latent dimension of 256 such that encodings are additionally much sparser in the latent space. We specifically set out to understand whether low dimensional spaces could encode TCRs well; for interpretability, potential TCR design, and downstream use-cases such as clustering. We believe with our extended TCR clustering benchmarking we have shown that such low-dimensional space can do this, while providing fast performance. Due to our quantification of smoothness and disentanglement we further show that this can provide interpretability of results, and offers potential for smooth generative capabilities from the latent space.

5. Once again, please provide a statistical test of your claim of superior clustering, and compare with another autoencoder baseline such as DeepTCR.

We would like to once again thank the reviewer for their time reading and providing thoughtful feedback on our manuscript. We hope to have addressed to reviewers concerns in our answers above, and with the extensions we have made to our manuscript.

REVIEWERS' COMMENTS

Reviewer #2 (Remarks to the Author):

The authors have addressed all my comments.

I would have appreciated a standard pypi package instead of a git, but it is also fine as is.

Reviewer #3 (Remarks to the Author):

I appreciate the clarifications and revisions made by the authors. I recommend the paper for publication, subject to the following revisions.

1. The authors have added the text “Such optimization requires a low dimensional space in which a continuous function is optimized”. This statement is not true, as others have demonstrated that antibody sequences can be optimized directly in one-hot encoded sequence space using gradients of an objective function (PMID: 31778140). Please correct this statement. In fact, this method worked better than using the encoding of CDR3 sequences by an autoencoder into a low dimensional space that was used for gradient optimization.
2. Please correct to say typically required. There are multiplexed methods that can resolve chain pairing without single-cell sequencing. “Although these chains occur in distinct pairs, single cell sequencing is required to resolve the pairing and thus the datasets of paired TCR chains are much smaller than datasets of independent α and β chains.”
3. FIG 1. Caption - “since the V gene usage can be encoded almost uniquely via CDR2” please add “it’s CDR2”
4. “Our VAE models on the physicochemical representations of TCRs generate a continuous space of PWMs.” This is misleading, as the VAE models include complex dependencies between residue representations that may not be adequately modeled by a reasonable number of PWMs that assume independence. Might you say “Our VAE models on the physicochemical representations of TCRs advance beyond what is possible with PWM

approaches.”?

5. Since you have not proposed a way to determine if the observed superior performance of your model is a result of chance, I would recommend that you change the text in your paper as follows - “The TCR-VALID curve is consistently above GLIPH2 [42] and deepTCR [13], a deep learning model with larger latent space (Fig.4c, left panel) indicating that it outperforms them ON THIS BENCHMARK. Further, we benchmarked TCR-VALID against recent general protein transformer-based models 265 [25] and TCR specific transformer models [10] (Fig.4c, left panel) that learn high dimensional 266 embeddings of TCR sequences (methods) and found THE CURVE OF OUR approach IS ABOVE these approaches.

SUBSEQUENT REVIEWER COMMENTS

Reviewer #2 (Remarks to the Author):

The authors have addressed all my comments.

I would have appreciated a standard pypi package instead of a git, but it is also fine as is.

We thank the reviewer for reading our revised manuscript.

Reviewer #3 (Remarks to the Author):

I appreciate the clarifications and revisions made by the authors. I recommend the paper for publication, subject to the following revisions.

We thank the reviewer for reading and recommending our revised manuscript for publication.

1. The authors have added the text “Such optimization requires a low dimensional space in which a continuous function is optimized”. This statement is not true, as others have demonstrated that antibody sequences can be optimized directly in one-hot encoded sequence space using gradients of an objective function (PMID: 31778140). Please correct this statement. In fact, this method worked better than using the encoding of CDR3 sequences by an autoencoder into a low dimensional space that was used for gradient optimization.

We thank the reviewer for pointing out a small oversight in our language here. We have updated the text to be specific to Bayesian optimization rather than “such optimization”. Bayesian Optimization is known to be more likely to be successful with few dimensions (typically less than 20), and we included a reference to a well-known and -cited review on the subject by Peter Frazier to aid the interested reader.

2. Please correct to say typically required. There are multiplexed methods that can resolve chain pairing without single-cell sequencing. “Although these chains occur in distinct pairs, single cell sequencing is required to resolve the pairing and thus the datasets of paired TCR chains are much smaller than datasets of independent α and β chains.”

We have made the suggested change.

3. FIG 1. Caption - “since the V gene usage can be encoded almost uniquely via CDR2” please add “it’s CDR2”

We have made the suggested change.

4. “Our VAE models on the physicochemical representations of TCRs generate a continuous

space of PWMs.” This is misleading, as the VAE models include complex dependencies between residue representations that may not be adequately modeled by a reasonable number of PWMs that assume independence. Might you say “Our VAE models on the physicochemical representations of TCRs advance beyond what is possible with PWM approaches.”?

We thank the reviewer for bringing this potential confusion to our attention. We have clarified this statement, to highlight our key point: that the output of the VAE is on a continuous space of physicochemical representations, and since these representations can be converted to PWMs (methods) they are themselves continuous and not discrete. We have changed this sentence to:

“Our VAE models on the physicochemical representations of TCRs generate a continuous space of physicochemical representations, and since these representations can be converted to PWMs (methods) these PWMs are themselves continuous and not discrete.”

5. Since you have not proposed a way to determine if the observed superior performance of your model is a result of chance, I would recommend that you change the text in your paper as follows - “The TCR-VALID curve is consistently above GLIPH2 [42] and deepTCR [13], a deep learning model with larger latent space (Fig.4c, left panel) indicating that it outperforms them ON THIS BENCHMARK. Further, we benchmarked TCR-VALID against recent general protein transformer-based models 265 [25] and TCR specific transformer models [10] (Fig.4c, left panel) that learn high dimensional 266 embeddings of TCR sequences (methods) and found THE CURVE OF OUR approach IS ABOVE these approaches.

We have made the suggested change.